# EGOHANDICL: EGOCENTRIC 3D HAND RECONSTRUCTION WITH IN-CONTEXT LEARNING

**Binzhu Xie**[1,2*], **Shi Qiu**[1,2*†], **Sicheng Zhang**[3], **Yinqiao Wang**[1,2], **Hao Xu**[1,2],
**Muzammal Naseer**[3], **Chi-Wing Fu**[1,2], **Pheng-Ann Heng**[1,2]
[1] Department of Computer Science and Engineering, The Chinese University of Hong Kong
[2] Institute of Medical Intelligence and XR, The Chinese University of Hong Kong
[3] Department of Computer Science, Khalifa University
{bzxie,shiqiu}@cse.cuhk.edu.hk

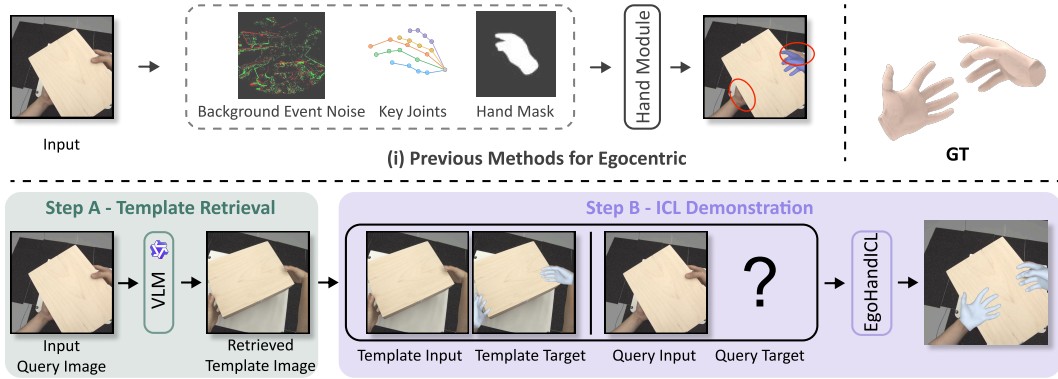

Figure 1: **Comparing EgoHandICL with previous methods.** (i) Prior works improve egocentric hand reconstruction by exploiting auxiliary cues (Prakash et al., 2024; Hara et al., 2025), thereby having limited capabilities in handling challenging scenarios with severe occlusions. (ii) EgoHandICL improves the precision through two key steps. **Step A:** We prompt vision-language models with egocentric cues to retrieve template images for the input query image. **Step B:** We construct ICL demonstrations by aligning the input-target pairs of both template and query images.

## ABSTRACT

Robust 3D hand reconstruction is challenging in egocentric vision due to depth ambiguity, self-occlusion, and complex hand-object interactions. Prior works attempt to mitigate the challenges by scaling up training data or incorporating auxiliary cues, often falling short of effectively handling unseen contexts. In this paper, we introduce **EgoHandICL**, the first in-context learning (ICL) framework for 3D hand reconstruction that achieves strong semantic alignment, visual consistency, and robustness under challenging egocentric conditions. Specifically, we develop (i) complementary exemplar retrieval strategies guided by vision–language models (VLMs), (ii) an ICL-tailored tokenizer that integrates multimodal context, and (iii) a Masked Autoencoders (MAE)-based architecture trained with 3D hand–guided geometric and perceptual objectives. By conducting comprehensive experiments on the ARCTIC and EgoExo4D benchmarks, our EgoHandICL consistently demonstrates significant improvements over state-of-the-art 3D hand reconstruction methods. We further show EgoHandICL's applicability by testing it on real-world egocentric cases and integrating it with EgoVLMs to enhance their hand–object interaction reasoning. Our code and data are available at: https://github.com/Nicous20/EgoHandICL.

---

[*]Equal contribution.
[†]Corresponding author.

# 1 INTRODUCTION

Reconstructing 3D hands from monocular RGB images has long been a core computer vision task, with a wide range of applications in extended reality (XR), human–computer interaction (HCI), robotics, *etc*. By leveraging transformer-based backbones trained on large-scale datasets to extract rich visual representations (Cui et al., 2023), recent methods, such as HaMeR (Pavlakos et al., 2024) and WiLoR (Potamias et al., 2025), achieve strong 3D hand reconstruction performance across various benchmarks. While effective, these models can still encounter practical challenges, such as depth ambiguity, self-occlusion, and domain shift, which often affect their robustness and generalization in real-world applications (Hong et al., 2022; Zhu et al., 2024; Cui et al., 2024). These challenges are further aggravated in egocentric settings, where severe occlusions, perspective distortions, and complex hand–object interactions are commonly present.

Previous works (Prakash et al., 2024; Hara et al., 2025) have also explored specialized solutions tailored for egocentric 3D hand reconstruction: as illustrated in Fig. 1-(i), current methods utilize auxiliary supervision cues that require additional annotations, but still fail to resolve severe occlusions or ambiguous hand-object interactions in egocentric views. These limitations highlight the urgent need for a flexible and generalizable framework that can adapt to diverse egocentric contexts with robustness. Although each egocentric scenario presents unique visual ambiguities, humans naturally resolve them by drawing on prior experience, multimodal context, and task-specific knowledge. This remarkable ability inherently aligns with the core concept of in-context learning (ICL) (Brown et al., 2020), which adapts to solve new problems by conditioning on a few relevant contextual examples. While ICL has achieved remarkable progress in language modeling (Dong et al., 2022; Ferber et al., 2024), recent studies have begun to apply this paradigm for vision tasks, where rapid problem adaptation and example-guided reasoning are equally critica (Zong et al., 2024). Given that ICL mirrors how humans exploit relevant cues to overcome ambiguity, it provides a natural paradigm for egocentric vision. Motivated by this inherent connection, we formulate a new framework, namely EgoHandICL, to advance egocentric 3D hand reconstruction by exploiting the strong reasoning capabilities of the in-context learning paradigm. As the first ICL-based approach for 3D hand reconstruction, we address the following two critical issues.

First, ICL's effectiveness highly depends on selecting relevant examples (Liu et al., 2022; Zhang et al., 2023; Rubin et al., 2022), yet egocentric scenarios make this selection particularly difficult. As shown in Fig. 1-(ii), we propose complementary retrieval strategies for exemplar template selection: (i) using four hand-involvement modes (left-hand, right-hand, two-hand, and non-hand) that broadly cover typical egocentric hand-related activities, we retrieve examples with similar hand involvement to ensure visual consistency between the query image and the matched templates; and (ii) we design interaction-specific prompts with a vision–language model (VLM) to retrieve adaptive templates conditioned on semantic context. Together, these strategies ensure that exemplars with strong semantic alignment and visual fidelity are exploited during in-context learning.

Second, unlike conventional ICL methods addressing single-modal mappings, 3D hand reconstruction requires bridging 2D visual inputs and 3D parametric outputs. To ensure structural consistency between queries and templates, we represent both inputs and outputs using a unified MANO parameterization (Romero et al., 2017), which yield semantically aligned input–output pairs that facilitate effective ICL. Moreover, we incorporate egocentric hand priors and multimodal cues to encode visual, textual, and structural context into a learnable token space, and adopt a Mask Autoencoders (MAE)-based architecture He et al. (2022) trained with 3D geometric and perceptual objectives. By conducting extensive experiments, we demonstrate that these designs enable our EgoHandICL framework to achieve more robust and generalizable egocentric hand reconstruction performance. In summary, our main contributions are threefold:

- We present the first in-context learning approach to 3D hand reconstruction, showcasing a promising way to tackle severe occlusions and diverse interactions in egocentric vision.
- We develop the EgoHandICL framework, which retrieves effective exemplars, tokenizes multimodal context, and learns through an MAE-style architecture for robust and generalizable 3D hand reconstruction.
- We present extensive experiments, demonstrating that our EgoHandICL achieves state-of-the-art performance on egocentric benchmarks and exhibits strong practicality in real-world scenarios, including self-captured data and enhancing hand–object interaction reasoning in VLMs.

## 2 RELATED WORK

**3D Hand Reconstruction.** Hand reconstruction has been extensively studied for a long time, where early works used to employ depth cameras to recover 3D hand joints and meshes (Ge et al., 2016; Oikonomidis et al., 2011; Tagliasacchi et al., 2015; Rogez et al., 2014; Simon et al., 2017; Sridhar et al., 2016; Sun et al., 2015; Tompson et al., 2014). A milestone is the introduction of MANO (Romero et al., 2017), a low-dimensional parametric hand model, which enables single-image 3D hand reconstruction by regressing associated posed shapes. Boukhayma et al. (2019) then demonstrates the applicability of MANO by using a CNN model and an articulated hand mesh. This learning-based approach has inspired following methods that either directly regress MANO parameters (Baek et al., 2019; Potamias et al., 2023; Baek et al., 2020) or predict hand vertices for improved image alignment (Kulon et al., 2019; Choi et al., 2020; Ge et al., 2019; Kulon et al., 2020). Beyond parametric regression, alternative strategies have been explored, including voxel-based (Iqbal et al., 2018; Moon & Lee, 2020) and vertex regression methods (Chen et al., 2021), with a few work further emphasizing efficiency and robustness of reconstruction (Chen et al., 2022; Park et al., 2022; Oh et al., 2023; Jiang et al., 2023). More recently, large-scale vision transformers pretrained on millions of images demonstrate that scaling both model capacity and data volume significantly enhances the generalization of hand reconstruction models to in-the-wild scenarios (Dong et al., 2024; Kim et al., 2023; Lin et al., 2021). In particular, HaMeR (Pavlakos et al., 2024) presents a powerful vision transformer-based (Vaswani et al., 2017) pipeline that utilizes patched image tokens to reconstruct the MANO parametric model, achieving state-of-the-art accuracy in both egocentric and in-the-wild settings. However, existing methods mainly focus on general reconstruction and lack robustness under challenging interactions and occlusions: *e.g.*, when hands cross and one is heavily obscured (Fig. 3), or when a hand blends into the background due to a black glove (Fig. 4). State-of-the-art approaches tend to miss hands, confuse left and right identities, or distort occluded regions. In contrast, our method leverages multimodal cues and acquires in-context knowledge from retrieved exemplars, achieving consistent and accurate hand reconstruction in these difficult egocentric scenarios.

**In-Context Learning in Vision.** Popularized by GPT-3 (Brown et al., 2020), in-context learning (ICL) allows models to adapt to new tasks by conditioning on a few input–output demonstrations without updating model parameters (Radford et al., 2021; Rubin et al., 2022; Xie et al., 2021). Inspired by the success of ICL in LLM exploration, researchers have also extended this paradigm to computer vision research, with representative efforts on segmentation (Li et al., 2024b), recognition (Zhang et al., 2025a), and multimodal understanding (Zong et al., 2024). However, applying ICL to 3D vision is more challenging because 3D data involves complex spatial and temporal structures that are not easily captured or represented by simple input–output pairs. Consequently, only a few studies have explored this area, such as PIC (Fang et al., 2023) for point cloud recognition, HiC (Liu et al., 2025) for human motion, and TrajICL (Fujii et al., 2025) for pedestrian trajectory forecasting, *etc*. Yet, none of them learns the contextual information needed to handle the large modality gap between 2D images and 3D meshes that exists in the reconstruction problem. To this end, we unify both modalities in the MANO parameter space and develop an ICL-specific tokenizer to effectively incorporate multimodal context. Moreover, within our EgoHandICL framework, the proposed VLM-guided exemplar retrieval strategies and MAE-driven reconstruction pipeline together facilitate semantic alignment and geometric consistency, enabling robust egocentric hand reconstruction even under severe occlusion and visual ambiguities.

## 3 METHOD

In this section, we first formulate in-context learning for egocentric 3D hand reconstruction (Sec. 3.1). Then, we present our EgoHandICL framework (Sec. 3.2), including template retrieval strategies that select exemplar samples, an ICL tokenizer that integrates visual, textual, and structural context, as well as the training and inference in practice. Finally, we introduce the specific losses that are used to train the EgoHandICL framework (Sec. 3.3).

### 3.1 MODELING IN-CONTEXT LEARNING IN EGOCENTRIC 3D HAND RECONSTRUCTION

**ICL Preliminaries.** In-context learning (ICL) typically models a task as few-shot inference, where a model $\mathcal{F}$ conditions on a small set of input–target exemplars and a query within a shared context.

Formally, given a query $x^{\mathrm{qry}}$ and a context set $\mathcal{C} = \{(x_i, y_i)\}_{i=1}^N$ of $N$ task-specific exemplar pairs (*i.e.*, "templates"), the model predicts $y^{\mathrm{qry}}$ by conditioning on both the query and the templates:

$$y^{\mathrm{qry}} = \mathcal{F}(x^{\mathrm{qry}}|\mathcal{C}). \tag{1}$$

**3D Hand Reconstruction.** Given an image $I \in \mathbb{R}^{H \times W \times 3}$, we aim to reconstruct the 3D mesh of each hand $i$ represented by the MANO model (Romero et al., 2017). We denote the set of MANO hand parameters as $\mathcal{M} = \{\Theta_i, \beta_i, \Phi_i\}_{i=1}^N$, where $\Theta_i \in \mathbb{R}^{15 \times 3}$ are the pose parameters, $\beta_i \in \mathbb{R}^{10}$ are the shape parameters, and $\Phi_i \in \mathbb{R}^3$ is the global orientation. Our objective is to learn a mapping:

$$\mathcal{M} = \mathcal{G}(I), \tag{2}$$

where $\mathcal{G}$ predicts the MANO parameters for all visible hands in the given image $I$. The estimated parameters $\mathcal{M}$ are then passed to the MANO decoder to obtain the corresponding 3D hand meshes.

**ICL for Egocentric 3D Hand Reconstruction.** Although we can realize 3D hand reconstruction by directly regressing MANO parameters from a single image, it becomes far more difficult in egocentric views, which often suffer from occlusions, complex hand–object interactions, and ambiguous viewpoints. To mitigate this, we leverage ICL with dedicated egocentric contextual exemplars, enabling example-based reasoning for accurate and robust egocentric 3D hand reconstruction. To fulfill the objective of Eq. 2 within the ICL formulation of Eq. 1, we proceed as follows.

Given a query image $I_{\mathrm{qry}}$, we first retrieve template images that are contextually related to the query. For each retrieved template image $I_{\mathrm{tpl}}$, we construct an input–target exemplar pair $(\tilde{\mathcal{M}}_{\mathrm{tpl}}, \mathcal{M}_{\mathrm{tpl}})$, aligning with both the objective of the 3D hand reconstruction task as well as the requirement of the ICL paradigm. Particularly, $\tilde{\mathcal{M}}_{\mathrm{tpl}}$ denotes coarse MANO parameters estimated by applying a trained reconstruction model (*e.g.*, HaMeR (Pavlakos et al., 2024) or WiLoR (Potamias et al., 2025)) to $I_{\mathrm{tpl}}$, as in Eq. 2; while $\mathcal{M}_{\mathrm{tpl}}$ denotes the retrieved template's ground-truth MANO parameters from the database. Then, we obtain a context set of egocentric exemplars $\mathcal{C}_{\mathcal{M}} = \{(\tilde{\mathcal{M}}_{\mathrm{tpl}}, \mathcal{M}_{\mathrm{tpl}})\}$. Finally, we estimate the coarse MANO parameters $\tilde{\mathcal{M}}_{\mathrm{qry}}$ of the query image $I_{\mathrm{qry}}$ with the same reconstruction model, and then refine it by conditioning on the exemplar set $\mathcal{C}_{\mathcal{M}}$ via the ICL paradigm (as Eq. 1):

$$\mathcal{M}_{\mathrm{qry}} = \mathcal{F}(\tilde{\mathcal{M}}_{\mathrm{qry}}|\mathcal{C}_{\mathcal{M}}). \tag{3}$$

This formulation establishes the foundation of applying ICL for egocentric 3D hand reconstruction.

## 3.2 EGOHANDICL FRAMEWORK

Fig. 2 illustrates the overall EgoHandICL framework. It is composed of three key components: template retrieval strategies, multimodal context tokenizer, and ICL learning and inference pipelines, where a VLM is prompted to retrieve the template mainly based on semantic descriptions derived from user-specified prompts regarding interactions, occlusions, and other egocentric cues.

**Template Retrieval.** As detailed in Sec. 3.1, a key step in our framework is retrieving a contextually relevant template image $I_{\mathrm{tpl}}$ for the input query image $I_{\mathrm{qry}}$. As shown in Fig. 2 (Part A), we propose two complementary retrieval strategies that exploit both visual and textual information for contextual adaptation within the ICL paradigm.

Specifically, we introduce *Pre-defined Visual Templates*, where the VLM classifies each image into one of four pre-defined egocentric hand-involvements: left hand-, right hand-, two hands-, and non hand-involvement types. This categorization covers common egocentric hand configurations, allowing us to retrieve visually consistent exemplars from the dataset. In addition, hand involvement alone is not always sufficient, as egocentric scenarios often include object interactions, occlusions, missing hands, *etc*. To address these issues, we further introduce *Adaptive Textual Templates*, where the VLM retrieves the template mainly based on semantic descriptions derived from user-specified prompts regarding interactions, occlusions, and other egocentric cues. This strategy enables more flexible and dynamic retrieval of semantically consistent exemplars, providing finer-grained contextual cues in addition to the basic pre-defined visual templates. In this work, we retrieve one template image per query image. Further implementation details are provided in Appendix B.

**ICL Tokenizer.** Once the template image is retrieved, we construct the ICL tokens by incorporating multimodal context from the template image $I_{\mathrm{tpl}}$ and the query image $I_{\mathrm{qry}}$. As shown in Fig. 2 (Part

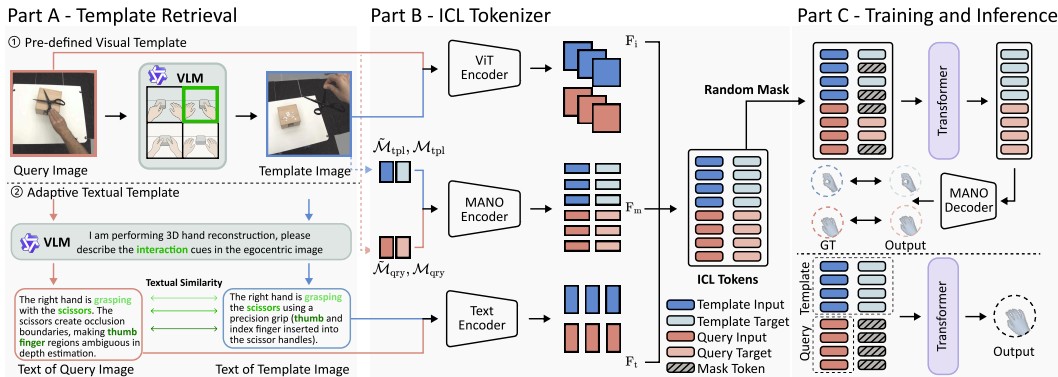

Figure 2: **Overview of our EgoHandICL framework. Part A:** Given a query image, we retrieve templates via two complementary strategies. *Pre-defined Visual Templates*: a VLM classifies the hand-involvement type and retrieves a template image of the same type. *Adaptive Textual Templates*: we prompt the VLM to generate semantic descriptions, and retrieve a template image given textual similarity. **Part B:** We encode image tokens $F_i$, structural tokens $F_m$, and text tokens $F_t$, respectively; and then apply cross-attention to tokenize four structured sets of ICL tokens. **Part C:** We follow a MAE-style design, where the template and query target tokens are partially masked to train the Transformer. In inference, the query target tokens are fully masked for the Transformer's prediction.

B), for both $I_{tpl}$ and $I_{qry}$, we estimate their coarse MANO parameters, collect their ground-truth MANO parameters, and feed these four sets of MANO parameters (*i.e.*, $\tilde{\mathcal{M}}_{tpl}, \tilde{\mathcal{M}}_{qry}, \mathcal{M}_{tpl}, \mathcal{M}_{qry}$) into a MANO encoder, generating the structural tokens $F_m$ that preserve 3D hand articulation and shape priors. In addition, we encode $I_{tpl}$ and $I_{qry}$, respectively, with a pretrained ViT encoder (Potamias et al., 2025), capturing appearance and spatial details and generating the image tokens $F_i$. Moreover, the VLM-generated semantic descriptions used for retrieval are embedded as the text tokens $F_t$ through a text encoder. Finally, we apply cross-attention to fuse the multi-modal *structural, image, and text tokens*, producing unified ICL tokens for context-aware reasoning. Specifically, after processing $I_{tpl}$ and $I_{qry}$, our ICL Tokenizer produces four sets of ICL tokens: $T_{tpl}^{in}$ (representing "input" of "template"), $T_{tpl}^{tar}$ (representing "target" of "template"), $T_{qry}^{in}$ (representing "input" of "query"), and $T_{qry}^{tar}$ (representing "target" of "query"). Together, these ICL tokens are utilized in the transformer-based reconstruction model as contextual exemplars.

**Masked Reconstruction for ICL.** Masked Autoencoders (MAE) (He et al., 2022) have been widely used in vision tasks, where part of the input tokens is randomly masked and a model is trained to reconstruct the missing ones. This architecture has proven effective for learning robust representations and handling incomplete or ambiguous visual signals. However, there is a key challenge when applying it to the ICL paradigm: during training, the model has access to the ground truths of both the template and the query samples; while in inference, the query target is unavailable, since it is exactly what we aim to infer. To tackle this, prior ICL methods for vision tasks (Bar et al., 2022; Wang et al., 2023; Fang et al., 2023) introduce masking into the target data during training: by partially masking the targets of both template and query samples, the model is trained to predict under incomplete supervision. As illustrated in Fig. 2 (Part C), our EgoHandICL employs a transformer trained for masked reconstruction over ICL tokens. During training, we randomly and partially mask the target tokens, *i.e.*, the ICL tokens of $T_{tpl}^{tar}$ and $T_{qry}^{tar}$, to simulate such incomplete supervision. In inference, $T_{qry}^{tar}$ is fully masked (unavailable), yet the trained model can decode (reconstruct) the target of query MANO parameters $\mathcal{M}_{qry}$ from the remaining ICL tokens of contextual exemplars. In summary, this MAE-driven design offers an effective training-inference architecture for exemplar-conditioned reasoning, enabling in-context hand reconstruction under challenging egocentric settings.

### 3.3 LOSS FUNCTION

We follow standard practices in parametric hand reconstruction and employ parameter-level ($\mathcal{L}_{mano}$) and vertex-level ($\mathcal{L}_V$) supervisions to train our EgoHandICL to reconstruct 3D hand meshes:

$$\mathcal{L}_{mano} = \left\|\Theta - \Theta^{gt}\right\|_2^2 + \left\|\beta - \beta^{gt}\right\|_2^2 + \left\|\Phi - \Phi^{gt}\right\|_2^2, \quad \mathcal{L}_V = \left\|V_{3D} - V_{3D}^{gt}\right\|_1. \quad (4)$$

While parameter and vertex supervisions provide essential geometric constraints, they remain insufficient in egocentric settings with lacking enough information only from the input image itself. Inspired by perceptual losses (Johnson et al., 2016) used for image restoration, we introduce a hand-specific 3D perceptual loss $\mathcal{L}_{3D}$, which aims to align high-level embeddings of predicted and ground-truth hand meshes. This loss enforces semantic consistency under occlusion and ambiguity, leading to reconstructions that are both geometrically accurate and perceptually realistic:

$$\mathcal{L}_{3D} = \left\| \phi(\mathcal{P}) - \phi(\mathcal{P}^{\text{gt}}) \right\|_2^2, \tag{5}$$

where $\mathcal{P}$ is the point cloud formed by the vertices (or joints) of the MANO-generated hand mesh, and $\phi(\cdot)$ is a pretrained 3D feature encoder. The overall loss $\mathcal{L}$ can be defined as:

$$\mathcal{L} = \lambda_m \mathcal{L}_{mano} + \lambda_v \mathcal{L}_V + \lambda_{3D} \mathcal{L}_{3D}, \tag{6}$$

where $\lambda_m$, $\lambda_v$, and $\lambda_{3D}$ are empirical weights. For datasets without MANO ground truth, *e.g.*, EgoExo4D (Chen et al., 2024), the loss applies 3D key-joint $J_{3D}$ constraints, weighted by $\lambda_j$:

$$\mathcal{L}_J = \left\| J_{3D} - J_{3D}^{\text{gt}} \right\|_1, \quad \mathcal{L} = \lambda_j \mathcal{L}_J + \lambda_{3D} \mathcal{L}_{3D}. \tag{7}$$

### 3.4 Implementation

For the retrieval process, we employ Qwen2.5-VL-72B-Instruct (Yang et al., 2024) as our VLM. For the ICL tokenizer, we use the same pretrained ViT backbone (Dosovitskiy et al., 2020) as in WiLoR, while the text encoder is Qwen-7B (Bai et al., 2023). The MANO encoder and decoder are implemented as MLPs, training together within the EgoHandICL framework. We implement the reconstruction model as a lightweight Transformer encoder. We also apply Uni3D-ti (Zhou et al., 2024) as the 3D feature encoder $\phi$ used in the loss $\mathcal{L}_{3D}$. All models are trained for 100 epochs with a learning rate of $1e-4$. The loss weights are set as $\lambda_m = 0.05$, $\lambda_v = 5.0$, $\lambda_j = 5.0$, and $\lambda_{3D} = 0.01$. The training is conducted on a single RTX 4090 GPU with a batch size of 64 and AdamW optimizer (Ilya, 2018). We use 4 A100 GPUs for data preprocessing and retrieval.

## 4 Experiments

### 4.1 Datasets and Evaluation Metrics

**Datasets.** We evaluate the EgoHandICL framework on two benchmarks: the ARCTIC dataset (Fan et al., 2023) provides high-quality MANO parameter annotations in controlled laboratory environments; and the EgoExo4D dataset (Grauman et al., 2024) consists of diverse egocentric videos with challenging hand-object interactions captured in realistic, unconstrained scenarios. These two datasets allow us to comprehensively evaluate the 3D hand reconstruction performance of EgoHandICL from the perspectives of hand mesh vertices (ARCTIC, 118.2K training / 16.9K testing samples) and hand skeleton joints (EgoExo4D, 17.3K training / 4.1K testing samples).

**Baselines.** We compare the performance of EgoHandICL with state-of-the-art 3D hand reconstruction baselines. For the ARCTIC dataset, we mainly evaluate against 3D hand mesh reconstruction methods, including HaMeR (Pavlakos et al., 2024), WiLoR (Potamias et al., 2025), Wild-Hand (Prakash et al., 2024) and HaWoR (Zhang et al., 2025b). For the EgoExo4D dataset, we primarily assess against 3D hand joint estimation approaches, including POTTER (Zheng et al., 2023), PCIE-EgoHandPose (Chen et al., 2024), and hand mesh reconstruction methods.

**Evaluation Metrics.** We evaluate egocentric hand reconstruction using both vertex-level and joint-level metrics, under two evaluation settings. In the general setting, metrics are computed for each detected hand, where cases with failed hand detection are excluded from evaluation. For ARCTIC, we report Procrustes-Aligned Mean Per Joint Position Error (P-MPJPE) and Procrustes-Aligned Mean Per Vertex Position Error (P-MPVPE), along with the fraction of vertices within 5mm and 15mm error thresholds (F@5, F@15). For EgoExo4D, we report the Mean Per Joint Position Error (MPJPE) and its Procrustes-aligned variant (P-MPJPE), together with F@10 and F@15 to measure the proportion of accurately reconstructed joints. To further capture the challenges of egocentric views while ensuring fair comparison under consistent detection conditions, we also evaluate under the bimanual setting, where only samples with both hands correctly detected are counted. In addition to P-MPVPE, we compute the Mean Relative Root Position Error (MRRPE) in this setting, to quantify the spatial consistency between the left and right hands (Fan et al., 2021; 2023; Moon et al., 2020). All metrics for both vertex-level and joint-level evaluations are reported in millimeters.

Table 1: **Quantitative results on the ARCTIC dataset.** We follow the standard evaluation protocol and report both the joint- and vertex-level metrics.

| Method | General Setting | | | | Bimanual Setting | |
|---|---|---|---|---|---|---|
| | P-MPJPE↓ | P-MPVPE↓ | F@5↑ | F@15↑ | P-MPVPE↓ | MRRPE↓ |
| HaMeR (Pavlakos et al., 2024) | 9.9 | 9.6 | 0.046 | 0.911 | 9.9 | 10.1 |
| WiLoR (Potamias et al., 2025) | 5.5 | 5.5 | 0.524 | 0.994 | 5.7 | 9.8 |
| WildHand (Prakash et al., 2024) | 5.8 | 5.6 | 0.746 | 0.928 | 4.9 | 7.1 |
| HaWoR (Zhang et al., 2025b) | 6.2 | 6.1 | 0.474 | 0.896 | 6.0 | 8.6 |
| **EgoHandICL (ours)** | 4.0 | 3.8 | 0.801 | 0.996 | 3.7 | 6.2 |

Table 2: **Quantitative results on the EgoExo4D dataset.** We follow the standard evaluation protocol and report the joint-level metrics.

| Method | General Setting | | | | Bimanual Setting | |
|---|---|---|---|---|---|---|
| | MPJPE↓ | P-MPJPE↓ | F@10↑ | F@15↑ | P-MPJPE↓ | MRRPE↓ |
| PCIE-EgoHandPose (Chen et al., 2024) | 25.5 | 8.5 | 0.544 | 0.910 | 8.2 | 130.9 |
| Potter (Zheng et al., 2023) | 28.9 | 11.1 | 0.491 | 0.910 | 10.3 | 148.9 |
| HaMeR (Pavlakos et al., 2024) | 30.1 | 11.2 | 0.453 | 0.883 | 10.6 | 361.2 |
| WiLoR (Potamias et al., 2025) | 31.1 | 12.5 | 0.528 | 0.905 | 11.0 | 378.0 |
| **EgoHandICL (ours)** | 21.1 | 7.7 | 0.789 | 0.935 | 7.5 | 110.9 |

## 4.2 EXPERIMENT ANALYSIS

**Experimental Results of 3D Hand Mesh Reconstruction.** We use the ARCTIC dataset for both joint- and vertex-level evaluations. As shown in Tab. 1, while WiLoR performs well when evaluating only detected hands in the general setting, its accuracy drops in bimanual cases, often failing to recover heavily occluded hands or confusing left/right identities (see the bottom case in Fig. 3). Compared to the second best, EgoHandICL consistently improves PA-MPVPE by 31.1% and 24.5% in the general and bimanual settings, respectively, and further reduces MRRPE by 12%. This indicates our finding that, with in-context learning, EgoHandICL is particularly effective at resolving egocentric occlusions and estimating spatial relations between the two hands. For the EgoExo4D dataset (Tab. 2), where MRRPE is substantially higher than in the ARCTIC dataset due to dynamic egocentric viewpoints, EgoHandICL again achieves notable gains with reduced errors, demonstrating strong spatial consistency between the two hands. Qualitative comparisons in Figs. 3 and 4 highlight that egocentric reconstruction involves not only single-hand accuracy but also maintaining reasonable relative geometry between the two hands in rapidly changing egocentric views. In general, our EgoHandICL consistently improves both visual accuracy and geometric fidelity across scenarios with egocentric occlusions and bimanual interaction, providing a more robust solution for real-world cases. Further discussions are provided in Appendix C.

Table 3: **In-context reasoning analysis across different hand-involvement types.** L, R, T, and N denote training on the left-hand, right-hand, two-hand, and non-hand involvement type sub-dataset, respectively. Results are tested on the ARCTIC under these four sub-dataset divisions.

| Type | Left Hand | | Right Hand | | Two Hands | | Non Hand | |
|---|---|---|---|---|---|---|---|---|
| Method | P-MPJPE↓ | P-MPVPE↓ | P-MPJPE↓ | P-MPVPE↓ | P-MPJPE↓ | P-MPVPE↓ | P-MPJPE↓ | P-MPVPE↓ |
| Proposed - L | 4.6 | 4.4 | 4.6 | 4.5 | 5.1 | 4.8 | 5.2 | 5.1 |
| Proposed - R | 4.8 | 4.5 | 4.1 | 4.4 | 5.0 | 4.8 | 5.2 | 5.2 |
| Proposed - T | 4.7 | 4.6 | 4.9 | 4.4 | 4.3 | 4.2 | 5.0 | 5.1 |
| Proposed - N | 4.9 | 4.7 | 5.3 | 5.1 | 5.5 | 5.1 | 5.0 | 5.0 |
| **Proposed - Full** | 4.5 | 4.3 | 4.0 | 3.9 | 3.9 | 3.7 | 4.7 | 4.5 |

**In-context Reasoning Analysis.** To verify the in-context reasoning abilities of the EgoHandICL framework, as well as the effectiveness of our categorization of pre-defined visual templates, we train variants of EgoHandICL on sub-datasets, each containing only a single hand-involvement type. As shown in Tab. 3, each variant performs the best when the evaluation condition matches the hand involvement type of its training templates. For example, Proposed-L achieves the lowest errors when testing on the sub-dataset of the left-hand involvement type. However, this configuration also

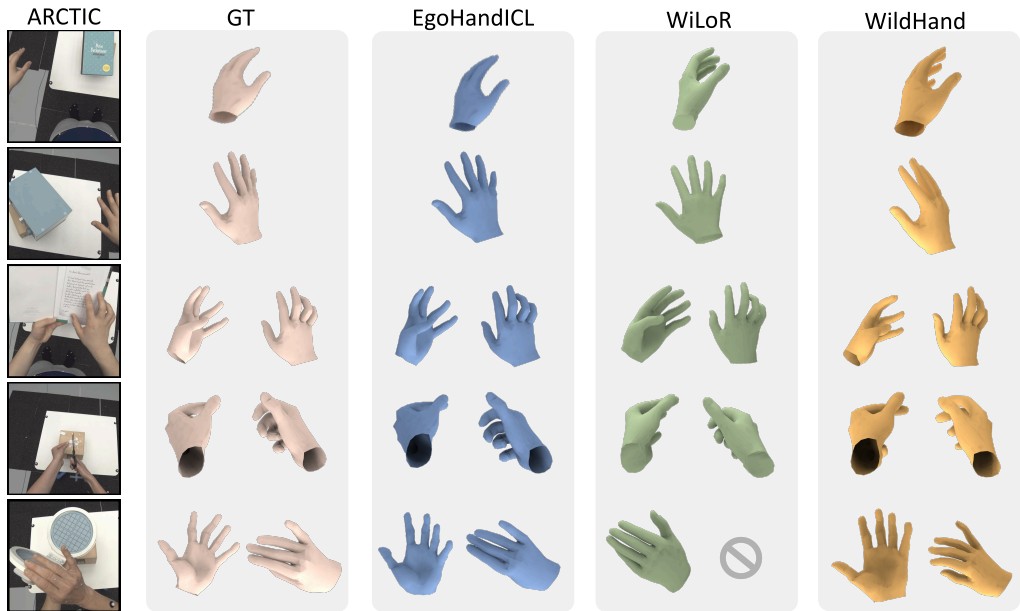

Figure 3: **Qualitative results on the ARCTIC dataset.** Note: In the bottom case, where the two hands cross and the left hand is severely occluded, WiLoR (Potamias et al., 2025) reconstructs only the right hand but mistakenly identifies it as the left.

introduces side effects: models trained only on a specific sub-dataset (*e.g.*, Proposed-N) generalize poorly to other types of samples (*e.g.*, the Two Hands sub-dataset). To this end, our Proposed-Full model, trained on all sub-datasets, achieves the best accuracy across all hand-involvement types, demonstrating the synergistic benefits of in-context learning for stronger generalization and effective adaptation to diverse scenarios.

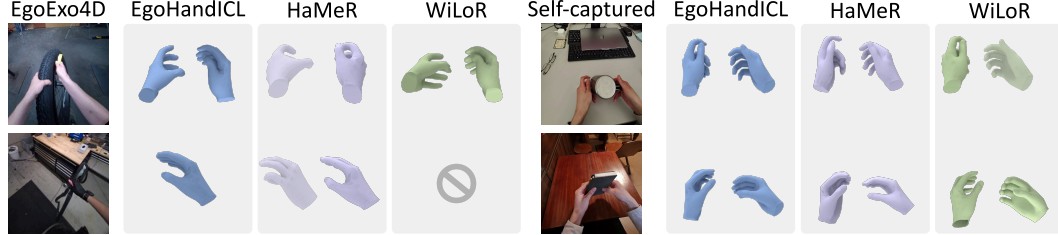

Figure 4: **Qualitative results on the EgoExo4D dataset (left) and self-captured cases (right)**. Note: In the bottom-left case with a single heavily occluded hand, HaMeR (Pavlakos et al., 2024) mistakenly reconstructs two hands, whereas WiLoR (Potamias et al., 2025) fails to reconstruct any.

**Different Prompts for Retrieval.** To assess the role of adaptive textual templates, we compare two prompting strategies for generating retrieval texts via the VLM (Tab. 4). When using only templates retrieved from the same hand-involvement type without any adaptive textual templates (*W/o.* Prompts), the model achieves relatively stable performance on F@5 and F@15. Introducing adaptive textual templates with descriptive prompts (*Des.* Prompts:

Table 4: **Comparison of different prompts for adaptive textual templates retrieval.** Results are tested on the ARCTIC dataset.

| Method | P-MPJPE↓ | P-MPVPE↓ | F@5↑ | F@15↑ |
|---|---|---|---|---|
| *W/o.* Prompts | 4.3 | 3.9 | 0.838 | 0.996 |
| *Des.* Prompts | 4.2 | 3.7 | 0.781 | 0.978 |
| *Reas.* Prompts | 3.9 | 3.7 | 0.766 | 0.975 |

*e.g.*, "*I am performing 3D hand reconstruction, please describe the interaction or occlusion details in the egocentric image.*") reduces reconstruction errors by providing explicit cues about occlusions. In particular, reasoning-style prompts (*Reas.* Prompts: *e.g.*, "*..., please provide guidance for han-*

*dling occlusions and complex interactions.*") further improve accuracy, as they encourage VLM to exploit richer contextual cues with semantic grounding. These results indicate that leveraging VLM's reasoning capabilities enables more effective and robust in-context learning for hand reconstruction. Our default setting balances this trade-off by applying the reasoning prompts under heavy occlusion and the description prompts in clearer scenarios.

## 4.3 ABLATION ANALYSIS

Table 5: **Comparison of coarse MANO prediction backbones.** Improvement denotes relative gains over the corresponding baseline performance on the ARCTIC dataset in Tab. 1.

| Method | P-MPJPE↓ | P-MPVPE↓ | F@5↑ |
|---|---|---|---|
| Ours *W.* HaMeR | 8.3 | 8.1 | 0.052 |
| **Improvement** | +16.1% | +10.4% | +13.4% |
| Ours *W.* WildHand | 5.1 | 4.9 | 0.823 |
| **Improvement** | +12.1% | +12.5% | +10.3% |
| Ours *W.* WiLoR | 4.0 | 3.8 | 0.805 |
| **Improvement** | +27.3% | +30.9% | +7.3% |

Table 6: **Comparison of different mask ratios for ICL tokens.** Results are tested on the ARCTIC dataset.

| Mask Ratio | P-MPJPE↓ | P-MPVPE↓ | F@5↑ | F@5↑ |
|---|---|---|---|---|
| 0.4 | 4.3 | 4.2 | 0.761 | 0.995 |
| 0.5 | 4.1 | 4.1 | 0.788 | 0.996 |
| 0.6 | 4.1 | 4.1 | 0.791 | 0.997 |
| 0.7 | 4.0 | 3.8 | 0.801 | 0.998 |
| 0.8 | 4.2 | 4.0 | 0.782 | 0.996 |

**Backbone for Coarse MANO Prediction.** Our EgoHandICL framework is designed to be independent of the choice of coarse MANO prediction backbone. To verify this, we use different MANO prediction backbones in our EgoHandICL framework. As shown in Tab. 5, EgoHandICL consistently and significantly improves over the corresponding baselines. These results indicate that the gains are attributed to the in-context learning paradigm itself, demonstrating the generalization and robustness of EgoHandICL regardless of the coarse MANO prediction backbone.

**Impact of Mask Ratio for ICL Tokens.** We evaluate the mask ratio over a broad range (40%–80%). As shown in Tab. 6, lower mask ratios moderately degrade performance, while a 70% mask ratio yields the best results. This suggests that in egocentric hand reconstruction, where occlusions and ambiguous interactions are common, masking a larger portion of ICL tokens encourages the model to exploit stronger contextual cues and employ deeper reasoning to infer hidden structures. The observation is consistent with the key insight of MAE (He et al., 2022), where a higher masking ratio enables the model to learn more informative latent features.

**3D Perceptual Loss.** We analyze the role of our introduced hand-specific 3D perceptual loss, which is designed to improve implicit hand representation alignment, complementing geometric parameter supervision. As Tab. 7 shows, incorporating our 3D perceptual supervision $\mathcal{L}_{3D}$ provides additional benefits over standard geometric regressions $\mathcal{L}_V$ and $\mathcal{L}_{mano}$. This indicates that hand-specific 3D perceptual loss $\mathcal{L}_{3D}$ adds complementary 3D cues that are not fully captured by $\mathcal{L}_V$ or $\mathcal{L}_{mano}$ alone.

Table 7: **Comparison of different loss items for the EgoHandICL training.** Results are tested on the ARCTIC dataset.

| Loss function | P-MPVPE↓ | F@5↑ | F@15↑ |
|---|---|---|---|
| $\mathcal{L}_V$ | 4.7 | 0.6 | 0.982 |
| $\mathcal{L}_V + \mathcal{L}_{mano}$ | 4.3 | 0.6 | 0.994 |
| $\mathcal{L}_V + \mathcal{L}_{3D}$ | 3.9 | 0.7 | 0.998 |
| $\mathcal{L}_V + \mathcal{L}_{3D} + \mathcal{L}_{mano}$ | 3.8 | 0.8 | 0.998 |

## 5 EXPLORING EGOVLMS WITH EGOHANDICL

Recent work has investigated egocentric video-language models (EgoVLMs) for hand-object interaction understanding. To explore whether EgoHandICL can facilitate this task, we evaluate on the EgoHOIBench (Xu et al., 2025) dataset with different EgoVLMs. As shown in Tab. 8, EgoGPT (Yang et al., 2025), which is built on LLaVA-OneVision (Li et al., 2024a) and fine-tuned on a wide range of egocentric data, underperforms the LLaVA-OneVision base model.

Table 8: **Comparison of EgoVLMs on hand-object interaction reasoning.**

| Model | avg.↑ | verb.↑ | none.↑ | action↑ |
|---|---|---|---|---|
| LLaVA-OneVision | 0.75 | 0.68 | 0.82 | 0.58 |
| + Proposed | 0.78 | 0.71 | 0.84 | 0.61 |
| EgoGPT | 0.71 | 0.66 | 0.76 | 0.46 |
| + Proposed | 0.76 | 0.70 | 0.90 | 0.61 |
| Qwen2.5-VL-7B-Instruct | 0.82 | 0.74 | 0.90 | 0.64 |
| + Proposed | 0.85 | 0.74 | 0.91 | 0.69 |

Figure 5: **EgoVLM's hand–object interaction reasoning with and without EgoHandICL.** By incorporating our hand reconstructions as visual prompts, hand-related actions in egocentric videos can be recognized reliably with finer details.

This suggests that excessive finetuning on egocentric data may bias EgoVLMs toward event-level understanding at the cost of fine-grained hand–object interaction reasoning. To mitigate this issue, we feed EgoHandICL's reconstruction outputs to the EgoVLM as additional visual prompts, consistently improving their hand-object interactions reasoning capabilities (an example shown in Fig. 5). This exploration demonstrates the practical value of EgoHandICL for downstream applications.

# 6 CONCLUSION AND FUTURE WORK

We present EgoHandICL, a novel framework for egocentric 3D hand reconstruction with in-context learning. Our method features three main components: (1) VLM-guided retrieval of exemplar templates; (2) an ICL tokenizer synergizing multimodal context; and (3) an MAE-based architecture trained with specialized 3D geometric and perceptual constraints. Extensive experiments on benchmarks and real-world applications validate the effectiveness of our approach. While effective, the computational cost of the VLM-based retrieval presents a limitation for real-time deployment. In the future, we plan to refine the training pipeline and generalize our EgoHandICL framework to broader egocentric settings, such as 3D hand pose tracking and hand–object reconstruction.

## ACKNOWLEDGMENTS

The work described in this paper was supported by the Research Grants Council of the Hong Kong Special Administrative Region, China, under Project T45-401/22-N; and in part by The Chinese University of Hong Kong, under Projects 4055212 and 6907743.

## ETHICS STATEMENT

We take ethical considerations as a priority in this work. All datasets used strictly follow their official licenses and community standards, and we adhere to the ICLR Code of Ethics throughout our research. Our study does not involve any new human subject data collection, sensitive personal information, or practices that raise ethical concerns. Further discussion of social impact and compliance details can be found in Appendix A.3.

## REPRODUCIBILITY STATEMENT

We place high importance on the reproducibility of our work. We provide an anonymous implementation of EgoHandICL, including training and evaluation scripts, at the following link: https://github.com/Nicous20/EgoHandICL.

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

APPENDIX

## A DISCUSSIONS

### A.1 LIMITATIONS

While EgoHandICL achieves strong performance on egocentric 3D hand reconstruction, it has several limitations. First, the reliance on VLM-based retrieval introduces a non-trivial computational overhead, which limits real-time deployment on resource-constrained devices. Second, the framework depends on high-quality template databases, and retrieval performance may degrade if templates lack diversity. Third, egocentric datasets with complete annotations remain limited. For example, the widely used EgoExo4D dataset provides only keypoint-level ground truth for hand pose estimation, lacking full MANO parameter supervision as available in ARCTIC. Relying solely on keypoint annotations restricts model generalization to complex real-world scenarios. Moreover, most current evaluations are performed with ground-truth bounding boxes, while robust hand detection itself is a major challenge in egocentric views. The absence of a fair end-to-end benchmark that jointly evaluates detection and reconstruction limits progress toward more realistic deployments.

### A.2 FUTURE WORK

Building on the identified limitations, several promising directions emerge for future work. First, to mitigate the computational overhead of VLM-retrieval, we plan to explore more efficient retrieval mechanisms, such as lightweight VLMs, approximate nearest-neighbor search, or retrieval-free strategies that embed contextual exemplars directly into the learning pipeline. Second, addressing the limited availability of fully annotated egocentric datasets, we emphasize the importance of constructing comprehensive 3D egocentric benchmarks with consistent annotations of hand pose, object pose, and body pose. Such datasets would enable more holistic evaluation of hand–object–body interactions and facilitate robust generalization in real-world scenarios. Third, to move beyond evaluations that rely on ground-truth bounding boxes, we advocate for the construction of fair end-to-end benchmarks that jointly assess detection and reconstruction in egocentric settings, better reflecting real deployment challenges. Finally, we will explore temporal extensions of EgoHandICL for continuous 3D hand tracking and broaden its scope to encompass richer egocentric tasks, such as joint hand–object reconstruction and gaze-conditioned interaction modeling, with a particular emphasis on lightweight designs suitable for AR/VR applications.

### A.3 SOCIAL IMPACT

Egocentric 3D hand reconstruction has promising applications in extended reality, assistive technologies, and robotics. However, it also introduces potential risks. First, continuous capture of egocentric data may raise privacy concerns, particularly regarding bystanders or sensitive environments. Second, the reconstructed hand data could be misused in surveillance or unauthorized tracking applications. To mitigate these risks, we emphasize that **all datasets used in this work comply with their official licenses and community standards, and we strictly adhere to ethical guidelines throughout data usage and research practices.**

### A.4 LLM USAGE

Large language models (LLMs) are used to moderately polish the paper writing. Specifically, LLMs are employed for improving grammar and formatting consistency of LaTeX content.

## B MORE DETAILS OF EGOHANDICL

### B.1 PRE-DEFINED VISUAL TEMPLATES

We first introduce the construction of *Pre-defined Visual Templates*, which categorize egocentric images into four canonical hand-involvement types: left-hand, right-hand, two-hand, and non-hand involvement.

**Prompt Design.** To enable a VLM to automatically categorize each image, we design explicit textual prompts describing the four hand-involvement types. Our prompt is:

```
SYSTEM_PROMPT = """
You are an image understanding agent. Your task is to analyze a
    first-person perspective image and classify the interaction status
    of the left and right hands.

Classification rules:
- 0: Only the left hand is interacting with an object
- 1: Only the right hand is interacting with an object
- 2: Both hands are interacting with an object
- 3: Neither hand is interacting with any object

Important notes:
- Occlusion caused by objects must be considered in determining whether
    a hand is interacting.
- Use your best reasoning based on the visual content to make this
    decision.

Your response must be a single number: one of [0, 1, 2, -1]. Do not
    include any explanation or additional text.
"""

USER_PROMPT = """
Please analyze the first-person perspective image and determine the
    interaction status of the hands.

Return only the correct label number based on the following:
- 0: Only the left hand is interacting
- 1: Only the right hand is interacting
- 2: Both hands are interacting
- 3: Neither hand is interacting

"""
```

This prompt is fixed across both training and inference to ensure consistency.

**Data Distribution.** Based on the above categorization, we partition the dataset into the four hand-involvement subsets. The overall distribution of samples across categories is summarized in Tab. B.1. This categorization allows us to balance template retrieval across common egocentric scenarios.

Table B.1: **Data distribution of pre-defined visual templates.** Each hand-involvement type is split into training, validation, and testing sets.

| Hand Involvement Type | Train | Val | Test | Total |
|---|---|---|---|---|
| Left Hand | 7,991 | 2,283 | 1,143 | 11,417 |
| Right Hand | 14,334 | 4,095 | 2,049 | 20,478 |
| Both Hand | 92,917 | 26,547 | 13,275 | 132,739 |
| None Hand | 2,996 | 856 | 428 | 4,280 |
| **Total** | **118,238** | **33,781** | **16,895** | **168,914** |

**Training Strategy.** During training-time data preprocessing, we perform the template-retrieval procedure three times for every candidate training sample. This repetition is used to verify the correctness of hand-involvement classification and ensure that each exemplar is assigned to the proper category. Only samples with consistent predictions across all three runs are kept in the exemplar pool. After this offline verification, the training process itself uses random sampling within

each validated subset to construct the exemplar set for each query image at every iteration. This stochastic sampling maintains exemplar diversity and mitigates overfitting.

**Inference Protocol.**  At inference time, we adopt the same VLM prompt to categorize the query image and retrieve templates from the training set accordingly.

## B.2  ADAPTIVE TEXTUAL TEMPLATES

**Prompt Design.**  We design two families of prompts with different semantic emphases:

- **Description-style prompts (Des. Prompts):** These provide explicit cues about the egocentric scene, encouraging the VLM to describe visible details. Our Prompts are:

```
Please analyze the egocentric image and return one concise sentence
    describing the visible hands and their interactions with
    surrounding objects for 3D hand reconstruction. The description
    should specify:

1. Which hand(s) are present (left hand, right hand, or both).
2. The type of interaction (e.g., grasping, holding, touching,
    pointing, resting).
3. The object involved, if any (e.g., a cup, phone, keyboard).
4. If no hand is clearly visible, explicitly state "no hand
    involvement."

Examples:
- "Left hand grasping a cup."
- "Right hand pointing at a phone."
- "Both hands holding a book."
- "No hand involvement."
```

The prompts is especially effective when the hand is clearly visible, helping the VLM retrieve templates with matching appearance and interaction patterns.

- **Reasoning-style prompts (Reas. Prompts):** These encourage the VLM to infer strategies for handling challenging cases. Our Prompts are:

```
SYSTEM_PROMPT = """
You are a reasoning agent specialized in egocentric hand-object
    interaction understanding. Your task is to analyze images captured
    from a first-person perspective and generate a caption that
    describes all hand and object interaction details relevant to 3D
    hand reconstruction.

Given a single input image, output one concise sentence that specifies
    which hand(s) (left/right/both) are visible, their pose, and their
    interaction with any objects. Be precise and descriptive about the
    hand-object interaction without adding explanations or speculations.
"""

USER_PROMPT = """
Please analyze the image and return one sentence describing the hands
    and their interactions with objects for hand reconstruction (e.g.,
    left hand grasping a cup, right hand resting on the table).
"""
```

The prompt is mostly useful for heavily occluded or ambiguous views, where direct visual cues are insufficient and deeper contextual reasoning is required.

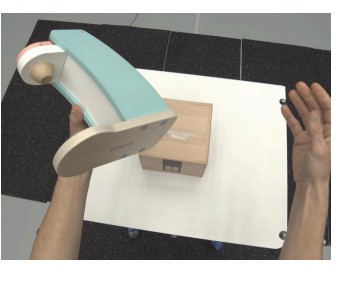

| | | |
|---|---|---|
| Hand-object Involvement Type: | Left | Right |
| Description Prompts: | Left hand holding a toy slide, right hand open and hovering nearby. | Right hands holding scissors and cutting paper on a wooden box. |
| Reasoning Prompts: | Left hand grasping a curved wooden toy with pastel-colored accents, right hand open and slightly raised above the table. | Right hand gripping black scissors with blades positioned over a wooden box, thumb and fingers engaged in cutting motion |

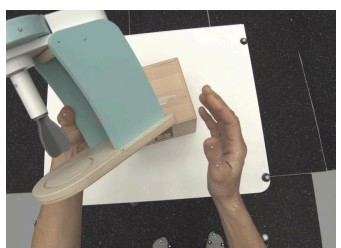
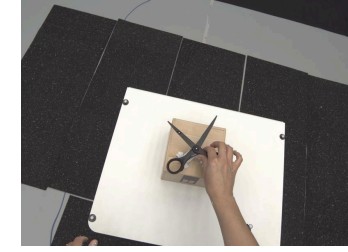

| | | |
|---|---|---|
| Hand-object Involvement Type: | Left | Right |
| Description Prompts: | Left hand touching the lid of a circular container. | Right hand holding black scissors above a white surface with a wooden box nearby. |
| Reasoning Prompts: | Left hand grasping a light blue and white toy structure with a wooden base, right hand open and positioned near the object as if preparing to assist or stabilize. | Right hand gripping black scissors with thumb through upper loop and fingers through lower loop, blades open and angled downward; left hand partially visible at bottom left. |

Figure B.1: **Query-Template Example 1.**

### B.3    EXAMPLES.

In Fig. B.1 and Fig. B.2, we illustrate examples of the two template retrieval strategies. During both training and inference, the retrieved images and the query image can serve as interchangeable query–template pairs, enabling flexible construction of in-context demonstrations.

## C    MORE DETAILS OF EXPERIMENTS

**Evaluation Settings.**    All baselines are fine-tuned on the same training splits, without incorporating any external data, ensuring a fully fair comparison. In the *General setting*, evaluation is performed on all hands detected within each image. However, egocentric scenarios often involve challenging cases such as partially invisible hands, severe self-occlusions, or multiple interacting hands from different persons. Under such conditions, different baseline models occasionally fail to detect all hands, leading to a small proportion of missing detections. Across our benchmarks, we observe that the proportion of such missing cases remains below 5%, and therefore the reported results in the general setting are still representative of overall performance.

**Bimanual setting.**    To provide a stricter and fairer comparison, we further adopt the *Bimanual setting*, where only samples with both hands correctly detected by each evaluated method are considered. This ensures that reconstruction accuracy is compared under consistent detection condi-

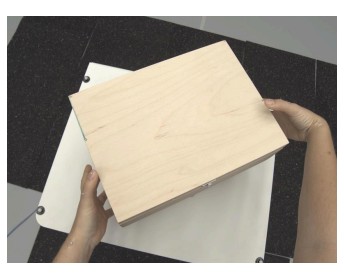

| | |
|---|---|
| **Hand-object Involvement Type:** | Both |
| **Description Prompts:** | Both hands lifting and tilting a wooden box. |
| **Reasoning Prompts:** | Both hands gripping a light wooden box — left hand supporting the bottom edge, right hand holding the top corner with fingers curled along the side. |

| | |
|---|---|
| **Hand-object Involvement Type:** | None |
| **Description Prompts:** | Both hands resting near a circular device on a white surface. |
| **Reasoning Prompts:** | Both hands hovering above the table with fingers slightly curled and marked for tracking, not in direct contact with the circular white object. |

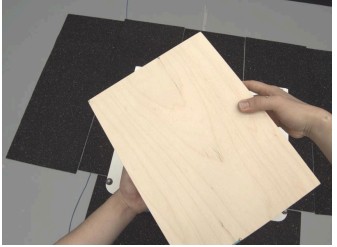
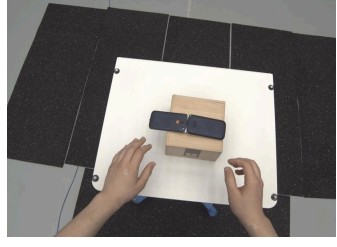

| | |
|---|---|
| **Hand-object Involvement Type:** | Both |
| **Description Prompts:** | Both hands holding a wooden board. |
| **Reasoning Prompts:** | Both hands holding a light wooden plank — left hand supporting the bottom edge with fingers extended, right hand gripping the top corner with thumb on front and fingers curled underneath. |

| | |
|---|---|
| **Hand-object Involvement Type:** | None |
| **Description Prompts:** | Two hands were suspended on the table. |
| **Reasoning Prompts:** | Both hands are poised just above the tabletop, lightly elevated with relaxed, curved fingers, while a foldable phone rests atop a small wooden box in the scene. |

Query Image with its templalte

Figure B.2: **Query-Template Example 2.**

tions across models, thereby eliminating potential biases caused by unequal detection performance. The resulting bimanual test set contains 12,041 and 3,802 samples on ARCTIC and EgoExo4D respectively, providing a robust basis for evaluating hand-to-hand spatial consistency and overall reconstruction reliability in dual-hand interactions.

**Performance across hand-involvement categories.** To better understand the behavior of different models under varying egocentric conditions, we further evaluate them separately on the four pre-defined hand-involvement types: left-hand, right-hand, two-hand, and no-hand. This breakdown allows us to quantify how well each model generalizes to different interaction settings, and to analyze whether certain models are more prone to performance degradation in specific categories.

Table C.1: **Performance of Model Zoos across different hand-involvement types.**

| Type | Left Hand | | Right Hand | | Two Hands | | Non Hand | |
|---|---|---|---|---|---|---|---|---|
| Method | P-MPJPE↓ | P-MPVPE↓ | P-MPJPE↓ | P-MPVPE↓ | P-MPJPE↓ | P-MPVPE↓ | P-MPJPE↓ | P-MPVPE↓ |
| HaMeR (Pavlakos et al., 2024) | 10.5 | 10.2 | 10.1 | 9.7 | 10.0 | 9.6 | 7.6 | 7.3 |
| WiLoR (Potamias et al., 2025) | 5.6 | 5.5 | 5.2 | 5.1 | 5.5 | 5.4 | 5.1 | 5.0 |
| WildHand (Prakash et al., 2024) | 5.9 | 5.8 | 5.6 | 5.4 | 5.3 | 5.1 | 5.2 | 5.0 |
| HaWoR (Zhang et al., 2025b) | 6.7 | 6.1 | 6.2 | 5.9 | 5.3 | 5.0 | 4.8 | 4.8 |
| Average | 7.2 | 6.9 | 6.8 | 6.5 | 6.5 | 6.3 | 5.7 | 5.5 |

As shown in Table C.1, the reconstruction errors vary across the four hand-involvement types. We observe that left-hand cases exhibit slightly higher errors (7.2/6.9) compared to right-hand cases (6.8/6.5). This asymmetry is consistent with prior observations that egocentric datasets are often right-hand dominant (e.g., ARCTIC (Fan et al., 2023), EgoExo4D (Chen et al., 2024)), leading to stronger representations for the right hand. Interestingly, two-hand scenarios achieve lower errors (6.5/6.3) than single-hand cases, suggesting that the presence of both hands provides additional geometric constraints that help the model resolve occlusions and perspective ambiguities. This trend contrasts with conventional baselines such as HaMeR (Pavlakos et al., 2024) and WiLoR (Potamias et al., 2025), which typically degrade in bimanual settings due to severe inter-hand occlusions. Finally, the lowest errors are obtained in the no-hand category (5.7/5.5), demonstrating the model's robustness in avoiding false positives when no hand is present.

# D    RETRIEVAL QUALITY AND OOD ANALYSIS.

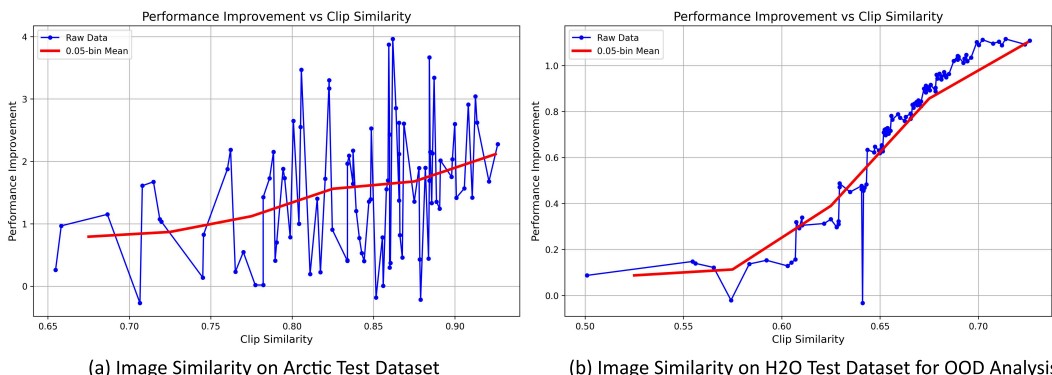

(a) Image Similarity on Arctic Test Dataset          (b) Image Similarity on H2O Test Dataset for OOD Analysis

Figure D.1: **Retrieval Quality and OOD Analysis.**

## D.1    RETRIEVAL QUALITY ANALYSIS

On the ARCTIC dataset, we assess the quality of the retrieved templates by computing the *cosine similarity* between the CLIP (Radford et al., 2021) image features of the query and its exemplar. We further correlate this similarity with the performance gain over the strongest baseline WiLoR (in P-MPVPE). As shown in Figure D.1 (a), over 90% of retrieved exemplars achieve a similarity above 0.7, indicating that our retrieval stage provides high-quality contextual matches. Moreover, higher similarity consistently leads to larger improvements over WiLoR, confirming that EgoHandICL effectively leverages high-quality exemplars for contextual refinement of 3D hand reconstruction.

## D.2 Out-of-Distribution (OOD) Analysis

We further analyze EgoHandICL under OOD conditions on the H2O dataset (Kwon et al., 2021), where retrieved templates are less aligned with the query image due to distribution shift. As shown in Figure D.1 (b), the cosine similarity between the query and its retrieved exemplar is predominantly concentrated in the range [0.6, 0.7], reflecting the difficulty of retrieval in OOD scenarios. Correspondingly, the absolute improvement over WiLoR is smaller than in the in-distribution case Figure D.1 (a), but EgoHandICL still consistently outperforms WiLoR across the observed similarity range, demonstrating robust generalization of our exemplar-driven ICL framework even when template quality is degraded.

## E  Failure Case

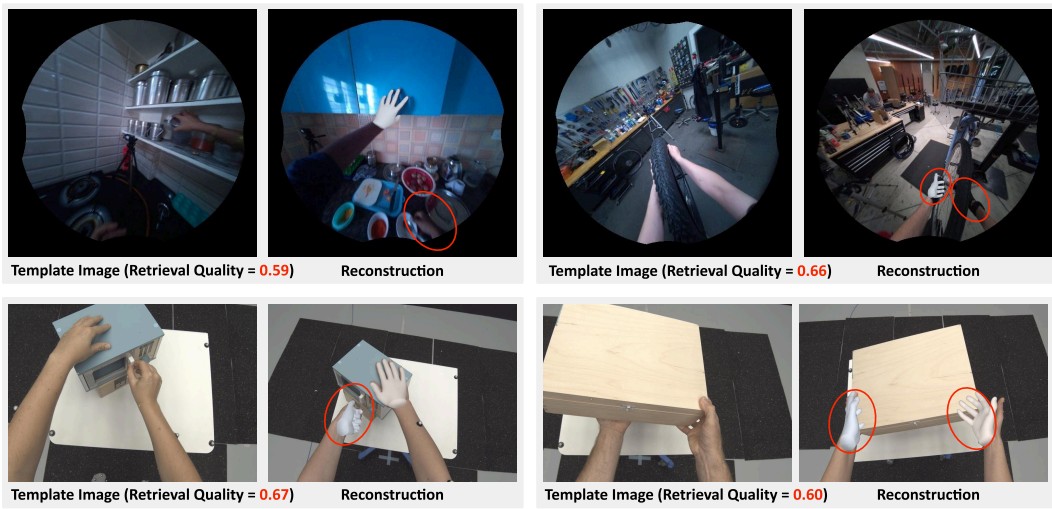

Figure E.1: **Failure Cases.** Failure cases under low-quality retrieval.

As shown in Figure E.1, EgoHandICL may fail when the retrieved exemplar is of low quality. Such low-quality retrieval typically arises under challenging visual conditions (e.g., severe occlusion, motion blur, or non-typical hand–object configurations), where the visual evidence becomes insufficient for accurate matching. In these scenarios, the retrieved exemplar offers weak contextual cues, leading to degraded reconstruction quality. Also, these are very challenging cases that existing prior methods can hardly handle.

