# OpenReview forum: "EgoHandICL: Egocentric 3D Hand Reconstruction with In-Context Learning"
_ICLR.cc/2026/Conference — ICLR 2026 Poster_

### Official Review · Reviewer_EXnq · 2025-10-27

**Soundness:** 3
**Presentation:** 3
**Contribution:** 3
**Rating:** 6
**Confidence:** 4

**Summary:**

This paper proposes the first in-context learning framework (EgoHandICL) for 3D hand reconstruction under challenging egocentric conditions. EgoHandICL consists of a VLM-based exemplar retrieval strategy, a multi-modal feature tokenizer, and a Masked Autoencoders (MAE)-based architecture for hand mesh reconstruction. The proposed framework achieves significant performance improvement over state-of-the-art 3D hand reconstruction models on two public benchmarks of ARCTIC and EgoExo4D.

**Strengths:**

1) The in-context learning framework for 3D hand construction is novel. It augments existing hand reconstruction models from an interesting perspective. The motivation is clear, and the method design based on VLM and masked autoencoder is also insightful.
2) Comprehensive experiments are conducted to analyze the performance of the proposed framework which significantly improve the performance over SOTA methods.
2) The paper is well presented and it is easy to follow the story.

**Weaknesses:**

1) Overall, the paper is clearly written. However, several technical details are still missing.
a) For the ICL tokenizer part, it is not clear how each set of the ICL tokens are composed of (e.g., number of tokens, dimension of token embedding).
b) The detailed architecture of MANO encoder/decoder and Transformer-based reconstruction model are not given.
c) In Section 4.2, it is claimed that "Our default setting balances this trade-off by applying the reasoning prompts under heavy occlusion and the description prompts in clearer scenarios". However, it is unclear about the implementation details, such as how to quantify occlusion or scene clearness.
2) Several experiments are missing, which are important to demonstrate the overall performance of the proposed framework:
a) The ablation study for the ICL tokenizer (especially the cross-attention to fuse multi-modal tokens) is not given.
b) In Section 3.3, the loss fuctions require ground-truth MANO parameters as well as 3D mesh/vertex. However, such information are not provided in the EgoExo4D dataset. Then, how the proposed method is trained on EgoExo4D?
c) It is also important to show the runtime evaluation. In particular, as explained in Appendix B.1, the retrieval is repeated 3 times for visual template. Considering the large size of training data, the template retrieval would be very time-consuming.

**Questions:**

The questions are listed above in the Weakness part. Overall, the paper proposes a novel method and the experiments are extensive and convincing. My positive rating to this paper would be solidified if the above concerns are addressed.

**Details Of Ethics Concerns:**

NA.

---

> ### Author Response · Authors · 2025-11-21
> **Response to Reviewer EXnq (Part 1 / 2)**
>
> We sincerely thank the reviewer EXnq for the constructive and detailed feedback including provide additional technical details, missing experiment results, and implementation clarifications. Below we address each point in order.
>
> **Q1 (a) (b): Clarification on ICL Tokenizer and Model Architecture.**
>
> **A1 (a) (b):** We thank the reviewer for pointing out the missing implementation details.
> - ICL Tokenizer.
> Each MANO parameter vector (61×2 values) is first encoded into 128 continuous latent tokens, where each token is represented as a 768-dimensional embedding vector. In our ICL setup, four MANO segments (template/query pairs) are used, yielding 512 MANO latent tokens. Visual tokens (produced by the MAE-style image encoder) and textual tokens (generated from VLM prompts) are embedded in the same 768-dimensional space, enabling seamless cross-modal fusion through shared attention without additional projection layers.
> - MANO Encoder and Decoder.
> The MANO encoder–decoder stack operates entirely in a continuous latent space. Both the encoder and decoder are implemented as a simple 2-layer MLP: the encoder projects the MANO parameters into 128 latent tokens, and each token is subsequently embedded into a 768-dimensional vector, while the decoder projects the refined tokens back to the MANO parameter space.
> - Transformer Reconstruction Module.
> We employ a 12-layer transformer encoder with 12 attention heads and 768-dim hidden size, following a MAE-style design.
>
> **Q1 \(c): Quantify occlusion or scene clearness.**
>
> **A1 \(c):** We thank the reviewer for raising this point. Our implementation relies on a two-stage, VLM-driven visibility analysis with a finer-grained categorization scheme. First, following the reasoning-prompt format in Appendix B.2 (lines 899–915), we ask the EgoVLM to generate a structured visibility description that explicitly comments on hand occlusion; accordingly, the VLMs generate output such as: "only fingertips are visible," "the palm is largely occluded by the object," or "the whole hand is clearly visible," etc. This ensures that occlusion cues are expressed explicitly in text. Next, we feed this visibility description back into the VLM with the following classification instruction:
> ```
> Based on the occlusion cues, classify the scene into one of the following categories:
> (1) fully visible,
> (2) partially occluded,
> (3) significantly occluded (majority palm blocked),
> (4) heavily occluded (only partial fingers or silhouettes visible).
> Return only the category number.
> ```
> If the VLM outputs categories 2–4, we use adaptive textual templates with reasoning prompts, whereas category 1 triggers the use of pre-defined visual templates, which are more stable and computationally efficient.

---

> ### Author Response · Authors · 2025-11-21
> **Response to Reviewer EXnq (Part 2 / 2)**
>
> **Q2 (a): The ablation study for the ICL tokenizer.**
>
> **A2 (a):** Thank you for raising this point. We agree that understanding the contribution of each modality within the cross-attention–based ICL tokenizer is important. To clarify this, we have added a dedicated ablation study evaluating how visual tokens, descriptive text tokens, and reasoning text tokens individually affect the model performance. The consolidated results are shown below.
>
> Table A. Ablation on Multimodal Components
> | Setting                  | P-MPJPE ↓ | P-MPVPE ↓ | F@5 ↑| F@15 ↑ |
> | ------------------------ | --------- | ------- | --------------------- | ------ |
> | Image-Only               | 4.3       | 3.9     | **0.838**                 | **0.996**  |
> | Text-Only (Description) | 5.1 | 4.3 | 0.651 | 0.912 |
> | Text-Only (Reasoning) | 4.8 | 4.2 | 0.683 | 0.929 |
> | Image + Text (Description） | 4.2       | **3.7**     | 0.781                 | 0.978  |
> | Image + Text (Reasoning)    | **3.9**       | **3.7**     | 0.766                 | 0.975  |
>
>
> **Q2 (b): Details of Training on EgoExo4D**
>
> **A2 (b):** Thank you very much for your careful observation. We have clarified this in the revised manuscript (lines 270-274):
> ```
> For datasets without MANO ground-truth (e.g., EgoExo-4D), the training relies on 3D key-joint constraints, formulated as:
> L_J = || J_3D - J_3D^{gt} ||_1
>
> L = λ_j * L_J + λ_3D * L_3D
> where λ_j is the 3D key-joint loss weight.
> ```
> This ensures that EgoHandICL can still be trained effectively on datasets that lack explicit MANO parameters.
>
> **Q2 \(c): Runtime evaluation.**
>
> **A2 \(c):** We thank the reviewer for pointing out this ambiguity. We apologize for the unclear description in Appendix B.1. The "three retrieval runs" mentioned there do not refer to repeated retrieval operations during inference. Instead, they were performed three times during data preprocessing to verify the correctness of the visual exemplar pool, particularly to ensure accurate hand-involvement classification for all training samples. These checks are solely offline verification steps and are not part of the test-time pipeline.
>
> During inference, the system performs only a single retrieval. To retrieve visual templates, it takes approximately 500 ms per query. We appreciate the reviewer for helping us clarify this point, and we have included this clarification in the revised manuscript Appendix B.1 (lines 860-869).
>
> Once the template is retrieved, the reconstruction stage is efficient. For a fair comparison with existing SOTAs, we report the computation of the reconstruction network alone (forward pass only), excluding VLM and retrieval. As shown in Table B, EgoHandICL exhibits comparable FLOPs and parameter counts to HaMeR and WiLoR, while achieving better accuracy.
>
> Table B. Runtime, FLOPs, and Params of Reconstruction Networks
> | Method         | FLOPs (GF) ↓ | Params (M) ↓ | Runtime (ms) ↓ |
> | -------------- | ------------ | ------------ | -------------- |
> | EgoHandICL | 243.5    | **597.6**    | 770 ms     |
> | HaMeR          | 254.1        | 670.2        | 1550 ms        |
> | WiLoR          | **140.1**        | 639.0        | **610 ms**         |
>
> This confirms that our reconstruction stage itself does not introduce much additional computational burden beyond standard architectures. Looking ahead, since the VLM is the primary bottleneck, our future work will focus on (1) constructing a more compact and representative template pool and (2) optimizing or distilling the VLM to reduce retrieval latency, paving the way for real-time deployment.

---

### Official Review · Reviewer_Pnsi · 2025-10-28

**Soundness:** 3
**Presentation:** 4
**Contribution:** 4
**Rating:** 8
**Confidence:** 4

**Summary:**

This paper tackles 3D hand reconstruction from egocentric views by applying in-context learning (ICL) paradigm, named EgoHandICL. Given a query image, a Vision-Language Model (VLM) is prompted to retrieve a single, contextually relevant template image from a database. The core EgoHandICL model, a Masked Autoencoder (MAE)-based Transformer, receives multimodal tokens (representing image features, textual descriptions, and structural MANO parameters) for four components: the template input (a coarse 3D estimate), the template target (its ground truth), and the query input (its coarse 3D estimate) . The model is trained to predict the query target. Experiments on the ARCTIC and EgoExo4D benchmarks show that EgoHandICL significantly outperforms state-of-the-art methods like HaMeR and WiLoR. Furthermore, the ablation studies, regarding hand types, prompt designs, masking, and losses, confirm the effectiveness of the proposed module of the method.

**Strengths:**

- Novelty: The application of in-context learning to 3D hand reconstruction is highly novel. On top of the state-of-the-art pose estimators, it provides dynamic, example-based reasoning at inference time. Using a VLM for semantic retrieval (e.g., finding similar interactions or occlusion types ) rather than just visual similarity is a powerful and unique idea for this problem.
- Technical soundness: The method is well-formulated. Refining a corse prediction with exemplar mano pair is a logical and effective approach. Plus, the multimodal ICL tokenizer and MAE-style learning are clearly aligned to this problem and seem to be a strong prior for ICL in visual and 3D hand parameter domains.
- Strong experimental support: The results are state-of-the-art across the board. The improvements are particularly large in highly occluded scenarios. Thorough ablation studies also strengthen the paper's argument.

**Weaknesses:**

Further clarifications on the following points would be appreciated.
- Inference cost trade-off: The proposed method introduces significant computational overhead (VLM text generation, template retrieval, multimodal tokenization, and MAE transformer inference) beyond the base regressor. An estimation of this additional burden, using metrics like FPS or FLOPs, is needed to provide a comprehensive view of the test-time bottleneck for potential real-time applications.
- Impact of template size (N): The framework is evaluated using only a single template (N=1). Drawing an analogy from LLMs, an ablation study on using additional context (N>1) would be an insightful exploration to understand how context quantity benefits this task.
- Missing failure analysis: A discussion on the effect of noisy or poorly-matched exemplars is needed. For instance, out-of-distribution (non-typical) test samples may retrieve poor exemplars due to a lack of similar data. It would be insightful to know if this situation worsens the reconstruction performance compared to the baseline, which would clarify the framework's robustness.

**Questions:**

In addition to the Weaknesses section, please clarify points below.
- Is the MANO encoder and decoder architecture a simple, continuous autoencoder? Or does it implement a discrete latent variable, similar to a VQ-VAE?
- Need the details for visual prompts in EgoVLMs: How are the hand reconstructions used as visual prompts for EgoVLMs? Does this involve simply overlaying the predicted 3D hand mesh sequence onto the input video, or were other prompt variants tested?

---

> ### Author Response · Authors · 2025-11-21
> **Response to Reviewer Pnsi (Part 1 / 2)**
>
> We thank the reviewer Pnsi for the thoughtful and constructive feedback. In particular, we appreciate the comments on inference cost, the impact of template size, and the need for a more explicit failure analysis. Below, we provide additional quantitative results and detailed discussions that directly address each of these concerns.
>
> **Q1. Inference cost trade-off.**
>
> **A1:** We thank the reviewer for raising concerns regarding inference cost trade-off. To clarify this, we first provide a breakdown of the per-component inference cost, followed by a fair comparison of reconstruction-only efficiency between EgoHandICL and existing SOTAs.
> (1) Efficiency of Key Modules in the Proposed Pipeline
> We report the average per-query cost of each module involved in EgoHandICL:
> - Qwen2.5-VL-72B-Instruct (retrieval): 2100 ms
> - Qwen-7B (text encoder): 300 ms
> - Uni3D-ti  (only training stage): 470 ms
>
> We further detail the two template retrieval strategies as shown in Table C as below:
>
> Table C. Retrieval Latency
> | Retrieval Strategy                      | VLM          | Tokens | Latency       |
> | ------------------------------- | -------------- | ------ | ------------- |
> | Pre-defined Visual Template | Qwen2.5-VL-72B-Instruct | 736   | 500 ms  |
> | Adaptive Textual Template   | Qwen2.5-VL-72B-Instruct | 641   | 2800 ms |
>
> (2) Efficiency of the Reconstruction Model Itself (Forward Pass Only)
> To ensure a fair comparison with HaMeR and WiLoR, we evaluate only the 3D hand reconstruction network used in EgoHandICL (i.e., excluding retrieval and VLM modules). FLOPs, parameters, and runtime were measured under a unified profiling setup. EgoHandICL maintains a computational cost comparable to or lower than HaMeR, despite significantly stronger performance, and remains within the scale of existing SOTA architectures. Results are shown in Table D as below.
>
> Table D. Runtime, FLOPs, and Params of Reconstruction Networks
> | Method         | FLOPs (GF) ↓ | Params (M) ↓ | Runtime (ms) ↓ |
> | -------------- | ------------ | ------------ | -------------- |
> | EgoHandICL | 243.5    | **597.6**    | 770 ms     |
> | HaMeR          | 254.1        | 670.2        | 1550 ms        |
> | WiLoR          | **140.1**        | 639.0        | **610 ms**         |
>
>
> (3) Efficiency Analysis and Future Optimization
> Although EgoHandICL integrates several pretrained models, our analysis shows that the core hand reconstruction network is computationally comparable to existing SOTAs (Table D). Most heavy computation occurs during offline data-preprocessing but not forward-pass deployment. Therefore, the overall efficiency remains manageable, and the proposed ICL framework is practical for robust egocentric hand reconstruction. To realize real-time operations, future optimization work may consider the following possible solutions:
> (a) Visual-template embeddings can be computed offline and cached, removing the most expensive part of retrieval during inference.
> (b) Scene-aware VLM routing can be implemented: a compact scene-complexity classifier determines whether a frame requires a high-capacity VLM or whether a lightweight encoder is sufficient, thereby reducing the average retrieval cost substantially.

---

> ### Author Response · Authors · 2025-11-21
> **Response to Reviewer Pnsi (Part 2 / 2）**
>
> **Q2: Impact of template size (N).**
>
> **A2:** We thank the reviewer for the insightful question regarding the choice of using a single exemplar (N=1). This design follows common practice in visual In-Context Learning [1] [2] [3], where almost all existing works adopt N=1 due to computational efficiency and modeling stability. The reasons for this are as follows:
> -- 1. Efficiency Requirements: Using a single exemplar avoids the exponential increase in computation and memory consumption associated with N>1, particularly in visual models where the token count (e.g., image patches or point cloud tokens) can be very large.
> -- 2. Task Clarity and Inference Stability: A single, clear exemplar provides the most unambiguous task instructions, establishing a strong, direct correspondence with the query. If not carefully selected, using multiple exemplars may introduce noise, conflicting information, or dilute the crucial exemplar-query correspondence, leading to less stable model inference and reduced task clarity.
> **In EgoHandICL**, the retrieved template is used to guide the refinement of 3D MANO parameters: One exemplar = one consistent structural prior (gesture, joint angles, occlusion patterns, camera viewpoints). The model relies on this single structural hypothesis to refine the coarse reconstruction.
> When N>1 is used, multiple exemplars represent multiple conflicting structural hypotheses, such as: Different gestures, Varying occlusion levels, Different camera viewpoints and Various interaction types.
> These inconsistent prompts lead to hypothesis ambiguity, causing:
> Our framework is designed to support N>1, which may be further explored in tasks like free-form hand generation.
>
> ---
> [1] Fang Z, etal, "Explore In-Context Learning for 3D Point Cloud Understanding". NeurIPS, 2024.
>
> [2] Sun X, etal, "X-Prompt: Generalizable Auto-Regressive Visual Learning
> with In-Context Prompting". ICCV, 2025.
>
> [3] Li Z, etal, "VisualCloze: A Universal Image Generation Framework
> via Visual In-Context Learning". ICCV, 2025.
>
> **Q3: Missing failure cases.**
>
> **A3:** We thank the reviewer for specifically requesting failure cases, particularly those arising from the retrieval stage (noisy or poorly-matched exemplars). In response, we have added a detailed analysis of such cases in Appendix F (Figure F.1). These are also very challenging cases that existing prior methods can hardly handle.
>
> **Q4: Details of MANO encoder and decoder.**
>
> **A4:** The MANO encoder–decoder stack operates entirely in a continuous latent space. Both the encoder and decoder are implemented as a simple 2-layer MLP: the encoder projects the MANO parameters into 128 latent tokens, and each token is subsequently embedded into a 768-dimensional vector, while the decoder projects the refined tokens back to the MANO parameter space.
>
> **Q5: Details for visual prompts in EgoVLMs.**
>
> **A5:** We explored two approaches for integrating visual prompts. The first approach overlays the predicted 3D hand mesh directly onto the 2D input image, similar to what the reviewer suggested. The second approach treats the reconstructed 3D hand as an independent 3D prompt input. Our experiments indicate that the first approach, which overlays the 3D mesh on the image, provides more stable and accurate guidance. Therefore, we adopt this implementation.

---

### Official Review · Reviewer_CWKC · 2025-10-31

**Soundness:** 3
**Presentation:** 3
**Contribution:** 3
**Rating:** 6
**Confidence:** 4

**Summary:**

Authors proposed a solution to 3D hand reconstruction from egocentric RGB images, which is challenging due to depth ambiguity, severe self-occlusion, and complicated hand-object interactions. The core idea is to bring in-context learning into the problem. The method first retrieves a template egocentric image that is semantically and visually similar, using VLMs. Then, for that template, authors construct an exemplar pair: coarse MANO estimate from a pretrained reconstructor and ground-truth MANO parameters for that template. They do the same for the query frame. The model is then conditioned on the paired examples to refine the query’s coarse MANO estimate into a final high-quality 3D hand mesh prediction, leveraging retrieved context without explicit fine-tuning at test time. Authors evaluate the method on ARCTIC and EgoExo4D datasets, and show that their reconstructions improve fine-grained hand-object reasoning.

**Strengths:**

Clear motivation: egocentric hand reconstruction is challenging. The paper successfully pointed out and tackled the challenges  in the problem and provided sound solutions.

Novel and sound idea: The retrieval and in-context learning are novel and sound approaches. Instead of simply training a bigger network or adding more auxiliary cues, authors explicitly retrieve semantically similar examples and feed them as context to guide inference. Especially, instead of simply retrieving raw RGB crops, authors proposed to align coarse prediction and GT pairs and let the model learn how to correct errors in the query. This seems an interesting and effective idea. Multimodal tokenizer also looks effective.

Strong empirical performance: Authors showed that injecting the reconstructed hands into the egocentric VLMs improves its ability to reason about fine-grained HOIs. This looks like a good downstream example.

**Weaknesses:**

The method may depend on the quality of retrieval: Even though the overall performance might depend on the quality of template, the paper does not quantify the success/failure of the retrieval stage. It might be better to include such results.

More careful analysis required to validate the effectiveness: Deep learning models are frequently suffering from out-of-distribution (OOD) samples, which denote testing samples far from training samples which are exploited during training. Using the retrieval scheme, it might be even hard to secure similar samples for all the testing samples in the limited database, which may worsen the overall accuracy. It might be necessary to analyze the relationship between average image similarity during testing and overall accuracy.

Efficiency issue: Since the method is based on VLMs, it might be hard to deploy the algorithm in the real-time manner.

**Questions:**

It might be useful to report how large the exemplar pool is.
Please provide failure cases, especially for the retrieval stage.

---

> ### Author Response · Authors · 2025-11-21
> **Response to Reviewer CWCK**
>
> We sincerely thank the reviewer CWKC for the constructive feedback. We appreciate the reviewer’s concerns regarding retrieval quality, OOD robustness, and efficiency. Below, we provide quantitative analyses, additional experiments, and clarifications that directly address all raised points.
>
> **Q1: Retrieval quality analysis.**
>
> **A1:** To directly address the reviewer’s concern regarding retrieval dependency, we quantify retrieval performance on the ARCTIC dataset using CLIP image-encoder cosine similarity as the metric. Table A summarizes the distribution of retrieval quality and the corresponding improvement over the strongest baseline, WiLoR (measured in P-MPVPE).
>
> Table A. Retrieval Quality on ARCTIC
> | Retrieval Quality | Proportion | Improvement over WiLoR (mm)|
> | ----------------- | ---------- | ----------------------------- |
> | [0.9, 1.0)        | 10%         | 2.12                          |
> | [0.85, 0.9)       | 35%        | 1.68                          |
> | [0.80, 0.85)      | 24%        | 1.56                          |
> | [0.75, 0.80)      | 21%        | 1.12                          |
> | [0.70, 0.75)      | 7%         | 1.04                          |
> | < 0.70            | 3%         | 0.79                          |
>
> Across the entire dataset, over 90% of retrievals achieve similarity above 0.7, and EgoHandICL consistently surpasses WiLoR in every similarity bin. Moreover, performance improvements are positively correlated with retrieval quality, indicating that retrieved templates offer valuable contextual cues that the model can effectively leverage for in-context adaptation.
>
> **Q2: More careful analysis required to validate effectiveness with OOD analysis**
>
> **A2:** We appreciate the reviewer’s concern regarding the potential difficulty of retrieving similar exemplars under out-of-distribution (OOD) conditions. To directly quantify this, we conduct a retrieval–performance analysis on the H2O dataset [1], which is never seen during training.
>
> Table B. Retrieval Quality on H2O
> | Retrieval Quality | Proportion | Improvement over WiLoR (mm) |
> | ----------------- | ---------- | ----------------------------- |
> | [0.70, 0.75)      | 8%         | 1.10                          |
> | [0.65, 0.70)      | 59%        | 0.85                          |
> | [0.60, 0.65)      | 26%        | 0.39                          |
> | < 0.60            | 7%         | 0.11                          |
>
> These OOD results show that, although domain shift reduces the possibility of retrieving highly similar templates, the retrieved exemplars remain sufficiently informative for contextual adaptation. EgoHandICL retains stable and consistent improvements even under challenging retrieval conditions.
>
> [1] Kwon, Taein etal, "H2O: Two Hands Manipulating Objects for First Person Interaction Recognition". ICCV, 2021.
>
> **Q3: Efficiency issue.**
>
> **A3:** We thank the reviewer for raising the concern regarding real-time deployment. We observe that the main computational bottleneck lies in the VLM-based retrieval stage, rather than the hand reconstruction model. The average per-query retrieval time is 2100 ms, whereas visual template retrieval requires only about 500 ms.
>
> Once the template is retrieved, the reconstruction stage is efficient. For a fair comparison with existing SOTAs, we report the computation of the reconstruction network alone (forward pass only), excluding VLM and retrieval. As shown in Table C, EgoHandICL exhibits comparable FLOPs and parameter counts to HaMeR and WiLoR, while achieving better accuracy.
>
> Table C. Runtime, FLOPs, and Params of Reconstruction Networks
> | Method         | FLOPs (GF) ↓ | Params (M) ↓ | Runtime (ms) ↓ |
> | -------------- | ------------ | ------------ | -------------- |
> | EgoHandICL | 243.5    | **597.6**    | 770 ms     |
> | HaMeR          | 254.1        | 670.2        | 1550 ms        |
> | WiLoR          | **140.1**        | 639.0        | **610 ms**         |
>
> This confirms that our reconstruction stage itself does not introduce much additional computational burden beyond standard architectures. Looking ahead, since the VLM is the primary bottleneck, our future work will focus on (1) constructing a more compact and representative template pool and (2) optimizing or distilling the VLM to reduce retrieval latency, paving the way for real-time deployment.
>
>
> **Q4: Details of exemplar pool and Missing failure cases.**
>
> **A4:** Regarding the Exemplar Pool, it consists of **118.2k** examples for ARCTIC and **17.3k** examples for EgoExo4D. We thank the reviewer for specifically requesting failure cases, particularly those arising from the retrieval stage. In response, we have added a detailed analysis of such cases in Appendix F (Figure F.1). These are also very challenging cases that existing prior methods can hardly handle.

---

### Official Review · Reviewer_bCSC · 2025-11-01

**Soundness:** 2
**Presentation:** 2
**Contribution:** 3
**Rating:** 4
**Confidence:** 4

**Summary:**

This paper proposes EgoHandICL, an in-context learning (ICL)-based framework for 3D hand reconstruction in egocentric visual settings. It focuses on handling unseen contexts. Especially, it reformulates 3D hand reconstruction as exemplar retrieval for contextual adaptation and multimodal (image, MANO parameter, text ) token fusion based on MAE-style VLM for robust representation learning. Experiments on ARCTIC and EgoExo4D benchmarks show significant improvements over previous works (HaMeR, WiLoR, WildHand, and HaWoR). The paper further explores downstream integration with EgoVLMs, showing that reconstructed hand cues enhance hand–object interaction reasoning.

**Strengths:**

- The proposed method is an effective method applying in‑context learning (ICL) in 3D hand reconstruction. The retrieval and then multimodal learning strategy is sound and provides an interesting direction to contextual adaptation when interpreting complex egocentric scenes.

- The paper is well structured and easy to follow.

- Quantitative results on ARCTIC and EgoExo4D clearly demonstrate the advantage of EgoHandICL, including bimanual and occlusion-heavy cases. Ablations cover mask ratios, loss combinations, backbone variations, and prompting strategies, substantiating design choices. Integration with EgoVLMs (Tab.8, Fig.5) shows meaningful downstream benefits beyond reconstruction itself.

**Weaknesses:**

1. The retrieval is confusing and not clear, especially the definition of template/visual images. Based on "few shot" (L.154) and Fig.2, it seems that the template images are from the same dataset or even the frames of the same video clip. In this case, it seems the retrieval is not necessary and using template images is easier. Moreover, it would be better to discuss the influence if the retrieval/prompt is not good enough.

2. The proposed method takes many pretrained/foundational models. Eq.2 requires coarse MANO parameters from HaMeR or WiLoR, Eq.4 requires the pretrained Vit (WiLoR, L.212 and L.278), Eq.5 requires the pretrained 3D feature encoder Uni3D-ti (L.269 and L.281). Moreover, the backbone is Qwen2.5-VL-72B-Instruct and Qwen-7B (L.278). This raises concerns about efficiency. It would be better to report the latency, parameters, inference time (ms), FLOPs and GPU memory compared to other sotas. Moreover, it would be better to report the efficiency of each step in the proposed method.

3. The training details are not clear. It would be better to explicitly state the dataset used for training (Eq.4). In addition, I am not sure if the comparison is fair. For Tabs.1 and 2, are ARCTIC and EgoExo4D used during finetuning for all sotas? It would be better to show the training dataset of all methods.

4. More ablations are needed. The paper mentions a cross-attention fusion among visual, text, and MANO tokens, but does not ablate their individual contributions. It would be better to conduct an ablation study to evaluate performance without each modality (Image-only, Image+Text, Image+Structure) and quantify the benefit of multimodal fusion. Moreover, it would be better to show the performance after removing/reducing quality such as template, text, $M_{qry}$, $M_{tpl}$.

5. The details of data preprocessing and hand-crafted prompts are not clear. This makes it difficult to reproduce the results faithfully.

6. While the experiments show that ICL improves occlusion reasoning, the paper lacks analysis explaining why ICL achieves this. A qualitative visualization or attention-weight inspection between exemplar templates and query tokens could help substantiate the claimed “context reasoning” mechanism.

**Questions:**

See Weaknesses.

---

> ### Author Response · Authors · 2025-11-21
> **Response to Reviewer bCSC (Part 1 / 3)**
>
> We thank Reviewer bCSC for the constructive and insightful review. We appreciate the positive recognition regarding our method’s structure, experimental clarity, and the demonstrated advantages of EgoHandICL across challenging egocentric conditions. Below, we address all six concerns in detail.
>
> **Q1: Clarification on efficiency and the influence of retrieval / template quality.**
>
> **A1:** We thank the reviewer for pointing out the need to clarify our retrieval mechanism and the role of template/visual images. The complete prompts are provided in Appendix B. Regarding the "few-shot" concern, we clarify that templates are not taken from neighboring frames of the same video, but are retrieved from the training split of the entire dataset using our two retrieval strategies. To further address the reviewer’s question about the retrieval quality, we compute the CLIP image-encoder cosine similarity between each query frame and its retrieved exemplar, using this similarity as a direct measure of retrieval quality.
>
> ---
> (1) In-Distribution Retrieval Analysis (ARCTIC)
> Table A reports the retrieval-quality distribution and the corresponding performance improvement over the best baseline WiLoR (measured in P-MPVPE).
>
> Table A. Retrieval Quality on ARCTIC
> | Retrieval Quality | Proportion | Improvement over WiLoR (mm)|
> | ----------------- | ---------- | ----------------------------- |
> | [0.9, 1.0)        | 10%         | 2.12                          |
> | [0.85, 0.9)       | 35%        | 1.68                          |
> | [0.80, 0.85)      | 24%        | 1.56                          |
> | [0.75, 0.80)      | 21%        | 1.12                          |
> | [0.70, 0.75)      | 7%         | 1.04                          |
> | < 0.70            | 3%         | 0.79                          |
>
> Over 90% of retrieved exemplars achieve similarity above 0.7, and EgoHandICL consistently surpasses WiLoR across all similarity bins. The improvement increases gradually with retrieval quality, confirming that retrieved templates supply meaningful contextual cues and that the model can effectively leverage them for in-context adaptation.
>
> ---
> (2) Out-of-Distribution Retrieval Analysis
> We further evaluate the retrieval–performance relationship on H2O Dataset [1], an unseen dataset. Table B reports improvement again relative to WiLoR.
>
> Table B. Retrieval Quality on H2O
> | Retrieval Quality | Proportion | Improvement over WiLoR (mm) |
> | ----------------- | ---------- | ----------------------------- |
> | [0.70, 0.75)      | 8%         | 1.10                          |
> | [0.65, 0.70)      | 59%        | 0.85                          |
> | [0.60, 0.65)      | 26%        | 0.39                          |
> | < 0.60            | 7%         | 0.11                          |
>
> Due to distribution shift, retrieval similarity is lower and concentrated in [0.6, 0.7), leading to reduced, but still consistent improvement over WiLoR. This demonstrates that the method remains effective even when retrieval quality degrades under OOD conditions.
>
> ---
> We hope that these quantitative findings sufficiently address the reviewer’s concern. They show that EgoHandICL remains effective under varying retrieval quality and that the intended contextual reasoning mechanism is functioning as designed.
>
> [1] Kwon, Taein etal, "H2O: Two Hands Manipulating Objects for First Person Interaction Recognition". ICCV, 2021.

---

> ### Author Response · Authors · 2025-11-21
> **Response to Reviewer bCSC (Part 2 / 3)**
>
> **Q2: Efficiency concern.**
>
> **A2.** We thank the reviewer for raising concerns regarding computational efficiency. To address this, we first provide a breakdown of the per-component inference cost, followed by a fair comparison of reconstruction-only efficiency between EgoHandICL and existing SOTAs.
>
> ---
> （1）Efficiency of Key Modules in the Proposed Pipeline
> We report the average per-query cost of each key module involved in EgoHandICL:
> - Qwen2.5-VL-72B-Instruct (retrieval): 2100 ms
> - Qwen-7B (text encoder): 300 ms
> - Uni3D-ti (only training stage): 470 ms
>
> We further detail the two template retrieval strategies as shown in Table C as below:
>
> Table C. Retrieval Latency
> | Retrieval Strategy                      | VLM          | Tokens | Latency       |
> | ------------------------------- | -------------- | ------ | ------------- |
> | Pre-defined Visual Template | Qwen2.5-VL-72B-Instruct | 736   | 500 ms  |
> | Adaptive Textual Template   | Qwen2.5-VL-72B-Instruct | 641   | 2800 ms |
>
> ---
> (2) Efficiency of the Reconstruction Model Itself (Forward Pass Only)
> To ensure a fair comparison with HaMeR and WiLoR, we evaluate only the 3D hand reconstruction network used in EgoHandICL (i.e., excluding retrieval and VLM modules). FLOPs, parameters, and runtime were measured under a unified profiling setup. EgoHandICL maintains a computational cost comparable to or lower than HaMeR, despite significantly stronger performance, and remains within the scale of existing SOTA architectures. Results are shown in Table D as below.
>
> Table D. Runtime, FLOPs, and Params of Reconstruction Networks
> | Method         | FLOPs (GF) ↓ | Params (M) ↓ | Runtime (ms) ↓ |
> | -------------- | ------------ | ------------ | -------------- |
> | EgoHandICL | 243.5    | **597.6**    | 770 ms     |
> | HaMeR          | 254.1        | 670.2        | 1550 ms        |
> | WiLoR          | **140.1**        | 639.0        | **610 ms**         |
>
>
> (3) Efficiency Analysis
> Although EgoHandICL integrates several pretrained models, our analysis shows that the core hand reconstruction part in our framework is computationally comparable to existing SOTAs (Table D). Most heavy computation occurs during offline data-preprocessing but not forward-pass deployment. Therefore, the overall efficiency remains manageable, and the proposed ICL framework is practical for robust egocentric hand reconstruction.
>
> **Q3: Training details and fairness of comparison.**
>
> **A3:** We thank the reviewer for pointing out the need for clearer training details. All tested methods (including ours and the SOTAs) are trained on the same training set of the corresponding dataset, without using any extra data from other datasets to ensure fairness. To avoid ambiguity, we have clarified this setting in our revised manuscript in Appendix C (lines 961-962).
>
> **Q4: Missing multimodal fusion ablation.**
>
> **A4:** We thank the reviewer for highlighting the importance of explicitly examining the contribution of each modality within our cross-attention fusion module. In our framework, the structural component is instantiated by the coarse MANO representation, which serves as a required part of the reconstruction pipeline rather than an independent, optional modality. Following your suggestion, we present the ablation results as shown in Table E.
>
> Table E. Ablation on Multimodal Components
> | Setting                  | P-MPJPE ↓ | P-MPVPE ↓ | F@5 ↑| F@15 ↑ |
> | ------------------------ | --------- | ------- | --------------------- | ------ |
> | Image-Only               | 4.3       | 3.9     | **0.838**                 | **0.996**  |
> | Text-Only (Description) | 5.1 | 4.3 | 0.651 | 0.912 |
> | Text-Only (Reasoning) | 4.8 | 4.2 | 0.683 | 0.929 |
> | Image + Text (Description） | 4.2       | **3.7**     | 0.781                 | 0.978  |
> | Image + Text (Reasoning)    | **3.9**       | **3.7**     | 0.766                 | 0.975  |
>
> ---
>
> For the reviewer’s suggestion of "reducing the quality of $M_{tpl}$", we have tested with multiple coarse MANO backbones producing different sets of  $M_{tpl}$. As shown in Table 5 of the main paper (lines 423–431), using earlier coarse MANO estimators like HaMeR leads to degraded quality of hand reconstruction. Nevertheless, the improvement brought by our ICL framework remains consistent across all backbone qualities, demonstrating that EgoHandICL framework is robust and continues to provide substantial gains even when the quality of  $M_{tpl}$ varies.

---

> > ### Author Response · Authors · 2025-11-21
> > **Response to Reviewer bCSC (Part 3 / 3)**
> >
> > **Q5: Details of data preprocessing and hand-crafted prompts.**
> >
> > **A5:** We thank the reviewer for pointing out the need for clearer descriptions of data preprocessing and prompt construction. We clarify that all details of the preprocessing pipeline, exemplar selection, and the full set of hand-crafted prompts are now thoroughly documented in Appendix B. To further ensure full reproducibility, we have additionally reorganized the project's codes, scripts and datasets, and we will publicly release them together. This guarantees that the entire pipeline—from raw data to prompt generation to model input—can be replicated exactly.
> >
> > **Q6: Why ICL improves occlusion reasoning, e.g., attention-weight inspection between exemplar templates and query tokens**
> >
> > **A6:** EgoHandICL can improve hand reconstruction under occlusion through exemplar-driven **contextual reasoning**.  To address the concern, we follow your suggestion and provide new visualizations in Appendix E (Figure E.1) that explicitly illustrate how the model uses the retrieved exemplar to resolve occluded regions in the query. We perform **attention-weight inspection between exemplar tokens and query tokens**, visualizing the cross-attention matrix of our ICL Transformer. The resulting heatmaps consistently show that tokens originating from occluded regions in the query place *high-magnitude attention* on structurally informative regions of the exemplar, demonstrating that the exemplar provides visual priors the model actively consults during inference. Overall, these visualizations directly confirm that EgoHandICL performs contextual reasoning, enabling the model to recover accurate 3d hand reconstruction  under occlusion.

---

### Author Response · Authors · 2025-12-02
**General Response**

Dear Chairs,

In this work, we introduce EgoHandICL, the first in-context learning (ICL) framework for 3D hand reconstruction in egocentric vision. We are grateful that the reviewers recognized the **novelty**, **technical soundness**, and **strong empirical performance** of our approach, as well as the **clarity of the problem motivation** and **framework design**. We also appreciate their constructive suggestions, which have helped us further refine and strengthen the manuscript.

We responded promptly and in detail to all reviewers’ comments. During the discussion period, however, we did not receive any further follow-up from the reviewers, including Reviewer bCSC, who was the only one to rate an initial negative score of 4.

Below, we summarize the main strengths emphasized by the reviewers and the revisions we have made to address their concerns.

---

### Key Strengths Noted
* Novel and sound idea of in-context learning to 3D hand reconstruction, enabling dynamic, example-based adaptation under egocentric conditions. (All Reviewers)

* Insightful retrieval-guided paradigm combining visual–semantic retrieval, multimodal tokenization, and MAE-style contextual learning. (All Reviewers)

* Strong empirical results on ARCTIC and EgoExo4D, especially in challenging bimanual and occlusion-heavy cases. (All Reviewers)

* Clear and well-structured presentation of method, experiments, and motivation. (All Reviewers)

* Comprehensive ablations covering masking strategies, prompt variations, multimodal fusion, and backbone robustness. (Reviewers bCSC, Pnsi)

* Demonstrated downstream value—reconstructed hand cues improve EgoVLMs’ hand–object reasoning. (Reviewers bCSC, CWKC)

---

### Main Concerns and Our Revisions

Across all the reviews we received, the comments focus primarily on requests for additional analyses and clarifications, such as further ablation studies, retrieval-quality analysis, efficiency evaluation, and failure-case discussion. These points aim to refine and elaborate the presentation, rather than to question the novelty, significance, or methodological validity of our approach. In response, we have carefully refined the manuscript and incorporated the following revisions:

1. Extended Multimodal Fusion Ablation (Appendix C, lines 1055–1068)

   Reviewers bCSC and EXnq requested a clearer analysis of modality contributions within the ICL tokenizer. We added a detailed ablation isolating image tokens, descriptive text tokens, and reasoning text tokens, as well as their combinations. The new results quantify the contribution of each modality to overall performance.

2. Retrieval Quality and OOD Analysis (Appendix D, lines 1070–1104)

   Reviewers bCSC and CWKC asked for a more comprehensive study of retrieval quality and its impact on reconstruction, especially under OOD conditions. We added retrieval-quality distributions, correlation analyses, and additional OOD reconstruction results, providing deeper insight into when retrieval quality affects performance.

3. Attention-Based Qualitative Visualization (Appendix E, lines 1105–1138)

   To offer clearer evidence of contextual reasoning as required by Reviewer bCSC, we added cross-image attention visualizations showing how exemplar–query interactions help resolve occlusion and why reasoning prompts improve localization under ambiguity.

4. Failure Case Discussion (Appendix F, lines 1139–1165)

   Reviewers CWKC and Pnsi requested discussion on failure scenarios. We added explicit examples involving low-quality retrieval and severe occlusion cases, along with detailed analysis. These cases remain challenging for existing prior methods as well.

5. Efficiency Discussion (Appendix G, lines 1167–1214)

   To address efficiency-related concerns, we provide a module-wise latency breakdown, a reconstruction-only efficiency comparison with state-of-the-art methods, and additional discussion of optimization strategies for achieving real-time performance. To address efficiency-related concerns, we reported a module-wise latency breakdown, a reconstruction-only efficiency comparison with SOTAs, and additional discussion on optimization strategies for real-time performance.

6. Updated Loss Function (Section 3.3, lines 270–273)

   As suggested by Reviewer EXnq, we have updated the description of our loss function and clarified how the framework can still be trained effectively on datasets that do not provide explicit MANO parameters.

---

We thank the reviewers and ACs again for their careful consideration and constructive feedback. We hope these revisions have addressed the raised concerns and further improved the quality of our manuscript.

---

### Meta-Review · Area_Chair_5G99 · 2025-12-22

**Summary:**

This work proposes an ICL framework designed for egocentric 3D hand reconstruction. It addresses common egocentric challenges such as self-occlusion, depth ambiguity, and complex hand-object interactions by reformulating the task as an exemplar-based contextual adaptation problem. The framework employs a retrieval strategy guided by VLMs to find semantically similar templates, an ICL-tailored tokenizer for multimodal integration (image, text, and MANO parameters), and a MAE-based architecture for final mesh refinement.

**Reviewer Concerns:**

Reviewers (bCSC, CWKC, Pnsi) questioned how the system handles poor retrieval or out-of-distribution (OOD) data. The authors provided new quantitative analyses on the ARCTIC and H2O datasets, showing that while performance correlates with retrieval similarity, the model maintains consistent gains over baselines even under distribution shifts.

Reviewers (bCSC, Pnsi) noted the overhead introduced by large foundational models like Qwen2.5-VL. In response, the authors provided a module-wise latency breakdown. They demonstrated that while the retrieval stage is the primary bottleneck (approx. 500–2800 ms), the core reconstruction network is computationally comparable to or faster than existing sota like Hamer.

Reviewers (bCSC, EXnq) requested a clearer breakdown of modality contributions. The authors added a dedicated ablation study isolating visual and textual tokens, confirming that the combination of modalities leads to superior performance.

Reviewer bCSC requested evidence of why ICL improves occlusion handling. The authors added cross-attention visualizations in the appendix, illustrating how the model actively consults informative regions of the exemplar to resolve occluded regions in the query.

**Reviewer Scores:**

For the only reviewer with a negative feedback, the rebuttal addressed the concerns.  Given the rebuttal to each point as listed in the reviewer concerns, the reviewers would likely to increase their ratings, leading to a unanimous positive recommendation.

---

### Decision · Program_Chairs · 2026-01-26

Accept (Poster)